# The conserved RNA-binding protein Seb1 promotes cotranscriptional ribosomal RNA processing by controlling RNA polymerase I progression

Maxime Duval[1], Carlo Yague-Sanz [1,4], Tomasz W. Turowski[2], Elisabeth Petfalski[3], David Tollervey [3] & François Bachand [1] ✉

Transcription by RNA polymerase I (RNAPI) represents most of the transcriptional activity in eukaryotic cells and is associated with the production of mature ribosomal RNA (rRNA). As several rRNA maturation steps are coupled to RNAPI transcription, the rate of RNAPI elongation directly influences processing of nascent pre-rRNA, and changes in RNAPI transcription rate can result in alternative rRNA processing pathways in response to growth conditions and stress. However, factors and mechanisms that control RNAPI progression by influencing transcription elongation rate remain poorly understood. We show here that the conserved fission yeast RNA-binding protein Seb1 associates with the RNAPI transcription machinery and promotes RNAPI pausing states along the rDNA. The overall faster progression of RNAPI at the rDNA in Seb1-deficient cells impaired cotranscriptional pre-rRNA processing and the production of mature rRNAs. Given that Seb1 also influences pre-mRNA processing by modulating RNAPII progression, our findings unveil Seb1 as a pause-promoting factor for RNA polymerases I and II to control cotranscriptional RNA processing.

DNA-directed RNA polymerases are a family of enzymes fundamental for the synthesis of RNA molecules in all living cells, a process known as transcription. In eukaryotic cells, RNA polymerase II (RNAPII) is responsible for the synthesis of messenger (m), small nucleolar (sno), most small nuclear (sn), and a variety of noncoding (nc) RNAs. The production of mature forms of these RNAs requires extensive processing and maturation events, many of which occur as the nascent transcript is being synthesized by RNAPII. The coupling between transcription and RNA processing is largely mediated by the carboxy-terminal domain (CTD) of the RNAPII catalytic subunit, which consists of heptad repeats with the consensus amino acid sequence $Y_1$-$S_2$-$P_3$-$T_4$-$S_5$-$P_6$-$S_7$. During the transcription cycle, CTD phosphorylation on Y1, S2, T4,

S5, and S7, is tightly controlled by various kinases and phosphatases[1,2], regulating the spatiotemporal recruitment of key factors that control cotranscriptional processing of nascent RNA. For instance, Ser5 phosphorylation on the CTD is essential during early RNA processing events such as capping of nascent mRNAs[3,4], while Ser2 phosphorylation acts in latter stages of cotranscriptional RNA processing, such as during mRNA 3' end maturation and transcription termination[5–7].

Genome-wide studies indicate that transcription elongation by RNAPII is a rather intermittent process, with frequent occurrences of pausing, backtracking, and arrest[8–10]. As several key steps of RNA maturation occur cotranscriptionally, variations in transcription elongation rates are therefore expected to have important

[1]RNA group, Department of Biochemistry & Functional Genomics, Université de Sherbrooke, Sherbrooke, QC, Canada. [2]Institute of Biochemistry and Biophysics, Polish Academy of Sciences, Warsaw, Poland. [3]Wellcome Centre for Cell Biology, University of Edinburgh, Edinburgh, UK. [4]Present address: URPHYM-GEMO, The University of Namur, 5000 Namur, Belgium. ✉e-mail: f.bachand@usherbrooke.ca

consequences on the efficiency of RNA processing. Indeed, evidence indicate that regulation of RNAPII elongation rates can influence cotranscriptional RNA processing events such as splicing[11–14] and alternative polyadenylation[15,16]. One RNAPII-associated factor that appears to influence RNA processing by controlling transcription elongation rate is the fission yeast Seb1 protein. At the amino acid level, *S. pombe* Seb1 shares homology with *S. cerevisiae* Nrd1 and human SCAF4/SCAF8 proteins, containing a CTD-interaction domain (CID) as well as a RNA recognition motif (RRM)[17]. In addition to preferential binding of the Seb1 CID to Ser2-phosphorylated versions of the RNAPII CTD, Seb1 also forms extensive physical contacts with the RNAPII core complex including regions in the vicinity of the RNA exit channel[18]. These Seb1-polymerase interactions allow cotranscriptional recruitment of Seb1 to RNAPII-transcribed genes, reaching a peak downstream of poly(A) sites where it was shown to associate with nascent pre-mRNAs[19,20]. Consistent with binding to nascent pre-mRNAs near the poly(A) signal, depletion of Seb1 causes global 3' UTR lengthening by impairing poly(A) site selection[19]. Although the exact mechanism by which Seb1 controls poly(A) site selection remains to be determined, reduction of transcription elongation rates attenuates Seb1-dependent 3' UTR length defects, suggesting that Seb1 controls RNA processing by modulating elongation kinetics[19]. Consistent with this view, NET-seq analysis in fission yeast revealed that Seb1 promotes long-live RNAPII pause states during transcription elongation[21].

Besides RNAPII, eukaryotic cells have two additional RNA polymerase complexes: RNAPI and RNAPIII that synthesize most rRNAs and tRNAs, respectively. Based on structural analyses and in vitro assays, all three eukaryotic RNA polymerases share a conserved basic architecture of the RNAP core, whereas peripheral factors are used to promote gene-specific transcription[22,23]. Yeast RNAPI consist of a 14-subunit complex[24] and is the most productive eukaryotic polymerase, contributing up to 60% of the total cellular transcriptional activity[25]. Transcription by RNAPI is functionally linked to ribosome biogenesis, as RNAPI has only a single genomic target, the multi-copy 9.1 kb-long 35 S rRNA gene. Transcription of the 35 S primary transcript by RNAPI is concurrently associated with pre-rRNA processing, which includes an extensive set of endonucleolytic cleavages and exonucleolytic trimming steps that involve >200 ribosome biogenesis factors[26]. Specifically, the primary rRNA transcript is processed into a (i) 20 S precursor of the mature 18 S rRNA that is part of the small ribosomal subunit and a (ii) 27 S precursor of the 5.8 S and 25 S rRNAs that make the large ribosomal subunit. Although these endonucleolytic cleavages can occur posttranscriptionally on the released 35 S primary transcript, metabolic labeling assays on short timescales indicate that most pre-rRNA processing occurs cotranscriptionally on nascent pre-rRNAs in yeast[27].

It is estimated that rapidly dividing yeast cells can generate new ribosomes at a rate of >2000/min, accounting for >75% of the total cellular transcription[25]. Given the substantial amount of cell resources invested in making ribosomes, it is not surprising that ribosome biosynthesis represents a crucial control point in the regulation of cell growth and division[28,29]. Regulation of RNAPI transcription is best characterized at the initiation step, where the activity of transcription initiation factors Rrn3 and SL1 is controlled in response to demands in protein synthesis[30,31]. In addition, growing evidence support that rRNA synthesis and pre-rRNA processing are also controlled during the elongation phase of RNAPI transcription[32–34]. Indeed, recent genomic approaches confirm that transcription elongation at the rDNA is not a continuous process during which RNAPI smoothly traverses the 35 S rRNA gene after initiation, but rather includes frequent RNAPI pausing events and/or regions with slower elongation rates[35,36]. Yet, cellular factors and molecular mechanisms that influence the transcription elongation rate of RNAPI remain poorly understood.

In this study, we show that the conserved fission yeast protein Seb1 associates with the RNAPI transcription machinery and is recruited to the 35 S rRNA gene. Seb1 deficiency results in reduced RNAPI density at the rDNA as well as decreases the cleavage efficiency of the 35 S primary transcript. Whereas the RNA-binding function of Seb1 is essential for its function in the regulation of RNAPI transcription, mutations that impair interaction of Seb1 with phosphorylated versions of the RNAPII CTD did not affect rRNA synthesis and pre-rRNA processing. Importantly, analysis of RNAPI distribution at the single nucleotide resolution revealed that a Seb1 deficiency leads to decreased RNAPI pausing/slow states during transcription elongation, consistent with the role of Seb1 in modulation of RNAPII elongation rate[19,21]. Our findings therefore unveil a role for Seb1 in the control of RNAPI transcription elongation and support a model where Seb1 serves as a general promoter of cotranscriptional RNA processing by decreasing transcription elongation rates.

## Results
### Seb1 is recruited to the rDNA and physically associates with the RNAPI transcription machinery and nascent pre-rRNA
We previously used ChIP-seq to map the binding profile of several pre-mRNA processing and transcription termination factors at the genome-wide level in fission yeast[7]. In addition to binding to mRNA genes, visual inspection of rDNA revealed Seb1 crosslinking throughout the rRNA gene (Fig. 1a). Importantly, recruitment of Seb1 to the rRNA gene was confirmed by ChIP-qPCR in multiple independent experiments (Fig. 1b, c). As shown for the association of Seb1 with protein-coding genes[19], crosslinking of Seb1 to rDNA was dependent on a functional RRM (Supplementary Fig. 1a, b), suggesting that interactions between Seb1 and nascent rRNA are required for robust recruitment of Seb1 to sites of rDNA transcription. We also detected recruitment of Dhp1 at the rDNA (Fig. 1a), consistent with the association of its homolog Rat1 in *S. cerevisiae*[37,38]. In contrast, no significant ChIP signal was detected for other mRNA processing factors (Rna14, Pcf11, Ysh1, and Cbp80) (Fig. 1a). As positive controls, two ribosome biogenesis factors, Cbf5 and Nop58, were analyzed and demonstrated extensive crosslinking at the rRNA gene (Fig. 1a). The generalized depletion of ChIP-seq signal in ITS1 and ITS2 regions of the rDNA gene appears to be the consequence of reduced DNA content for those regions in the starting chromatin extracts. Based on our ChIP assays, we conclude that Seb1 is recruited to sites of rRNA transcription.

Next, we examined the protein interaction network of Seb1 to assess for physical interactions with the RNAPI transcription machinery using two independent, yet complementary approaches: affinity purification coupled to mass spectrometry (AP-MS) and proximity-dependent biotinylation coupled to MS (PDB-MS). The AP-MS analysis of Seb1 was previously described using a fission yeast strain that expressed a HTP (6xHis-TEV-2xProA)-tagged version of Seb1 from its endogenous chromosomal locus[19]. For the PDB-MS analysis, we generated a strain that expresses Seb1 fused to the TurboID biotin ligase, as we have recently described[39]. In total, the AP-MS and PDB-MS approaches identified 1048[19] and 443 Seb1-associated proteins, respectively (Fig. 1d and Supplementary Data 1). Importantly, 268 proteins were common to both the AP-MS and PDB-MS assays (Fig. 1d and Supplementary Data 2). Gene ontology (GO) analysis of these 268 Seb1-associated proteins revealed two main biological processes among the top-10 GO categories: (1) RNAPI-dependent rRNA transcription/maturation and (2) RNAPII-dependent RNA 3' end processing (Fig. 1e). In total, 51 (19%) of the 268 Seb1-associated proteins have functions associated with RNAPI transcription and ribosome biogenesis (Fig. 1f), including proteins connected to the RNAPI core complex (Rpa34, Nuc1, Rpa49, Rrn3, Rpa43, Rpb10, and Rpc40) and positive regulators of RNAPI transcription (Utp10, SPBC28F2.11, SPCC4b3.08, and Prh1).

We also examined a previously described map of Seb1-RNA interactions in vivo[19] determined by UV-induced RNA-protein cross-linking and analysis of cDNA (CRAC) to assess for direct association of Seb1

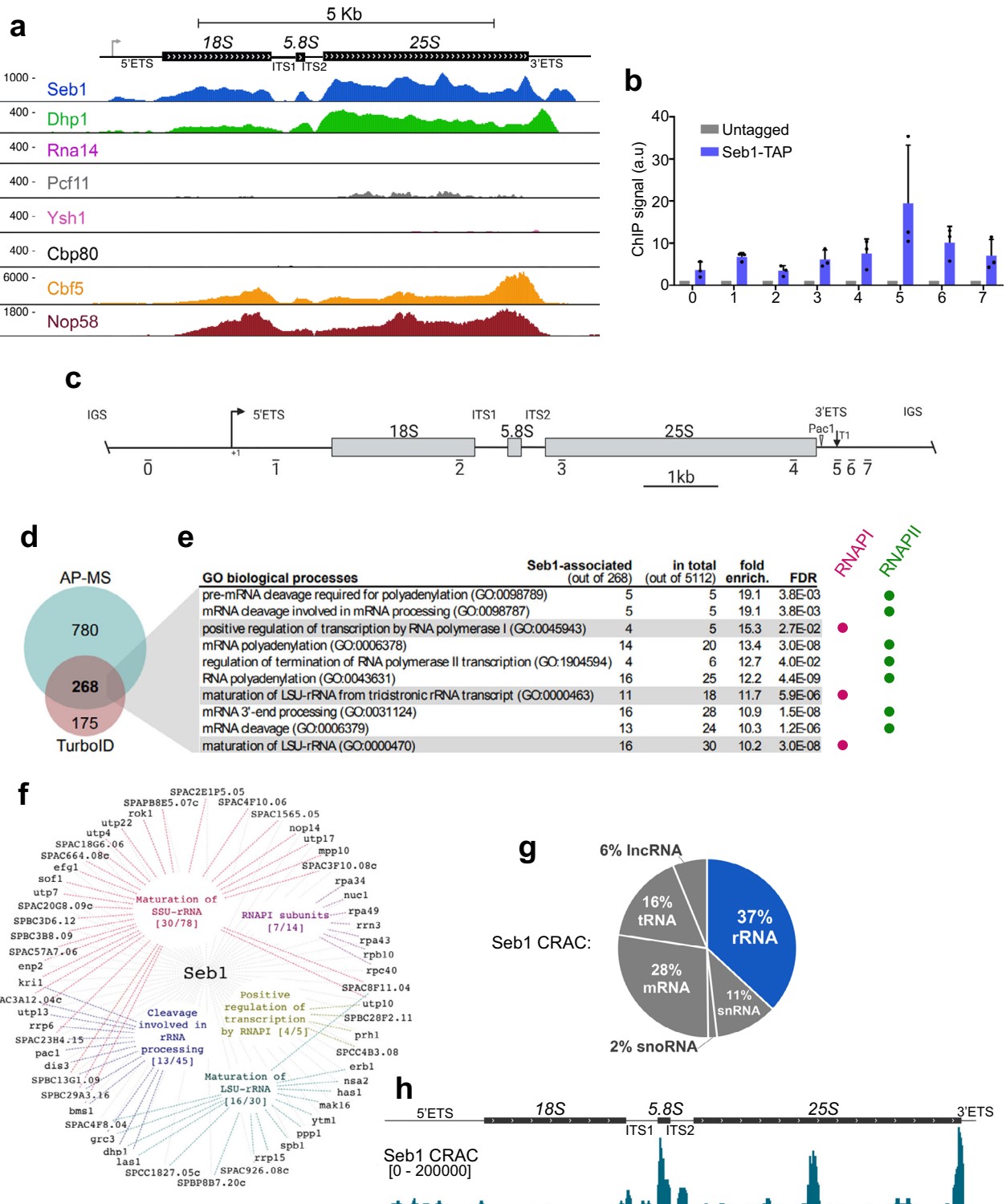

**Fig. 1 | Seb1 is recruited to the rDNA and physically associates with the RNAPI transcription machinery. a** Normalized ChIP-seq signal of Seb1-HTP and of the indicated proteins involved in mRNA processing (Rna14, Pcf11, Ysh1, and Cbp80), transcription termination (Dhp1), and ribosome biogenesis (Cbf5 and Nop58) on one representative rDNA repeat. The coverage is expressed in thousands reads mapped and averaged over two biological replicates. **b** ChIP-qPCR analysis of Seb1-TAP at the rDNA locus relative to an untagged control strain. a.u.: arbitrary units. Data and error bars represent the mean and standard deviation of *N* = 3 independent experiments. **c** Schematic of the rRNA gene. The bars under the rDNA indicate the positions of the PCR products used in the ChIP-qPCR analysis. **d** Venn diagram showing 268 Seb1-associated proteins identified independently in both AP-MS of

Seb1-HTP and proximity-dependent biotinylation of Seb1-TurboID analyses. **e** Top-10 most enriched gene ontology (GO) terms by biological process (fold enrichment calculated using amigo2) among the 268 Seb1-associated proteins. Terms related to RNAPI and RNAPII transcription are indicated by purple and green circles, respectively. FDR = false discovery rate. **f** 51 high-confidence Seb1-associated proteins that possess functions related to RNAPI transcription and ribosome biogenesis (from the 268 identified interactions). **g** Distribution of Seb1 CRAC reads[19] across the indicated categories of genes. **h** Seb1 CRAC read distribution on one representative rDNA repeat in a wild-type strain averaged over two biological replicates.

with rRNA. By including multi-mapped reads in the analysis to account for the multi-copy nature of rRNA genes, a breakdown of hits from two independent CRAC experiments revealed that nearly 37% of Seb1 binding mapped to rRNA genes (Fig. 1g). mRNA genes were the second most abundant gene type (nearly 28%), consistent with the role of Seb1 in mRNA poly(A) site selection[19] and association with the cleavage and polyadenylation complex (Fig. 1d, e). Interestingly, the distribution of Seb1 CRAC hits in the rRNA gene revealed strongest cross-linking in the internal transcribed spacer 1 (ITS1)−5.8S-ITS2 region, inside the 25 S, and at the 3' end of the rRNA gene, including the 3'-external transcribed spacer (3'ETS) (Fig. 1h). Importantly, Seb1-RNA interactions to those rRNA regions are not the consequences of non-specific associations, as they were either completely absent (ITS1, ITS2, 3'ETS) or largely depleted (5.8 S) in control CRAC experiments (Supplementary Fig. 1c). The direct binding of Seb1 to RNAs transcribed from the 5'ETS, ITS1, ITS2, and 3'ETS regions also support binding to pre-rRNAs. Collectively, our ChIP, protein interaction assays, and CRAC analysis indicate that a fraction of Seb1 is associated with the RNAPI transcription machinery at the rDNA where it binds to pre-rRNAs.

## RNAPI density at the rDNA is impaired when Seb1 is absent

To determine whether Seb1 functions in rRNA transcription, we analyzed RNAPI occupancy (via ChIP assays of myc-tagged Rpa2, the second-largest subunit of the core RNAPI complex) at the rDNA locus in wild-type cells and in a $P_{nmt1}$-seb1 conditional strain. The $P_{nmt1}$-seb1 strain was previously characterized[19] and represses seb1 expression after addition of thiamine to the growth medium (Supplementary Fig. 2a). As expected, ChIP assays of Rpa2-myc in wild-type fission yeast detected strong RNAPI binding along the transcribed regions of the rRNA gene as compared to an untagged control strain (Figs. 2a, b, gray versus blue bars). The decrease in Rpa2-myc occupancy downstream of the 3'-ETS (Fig. 2b, compare regions 5-6 to region 4) is consistent with efficient termination of RNAPI transcription in yeast[37,38]. Notably, cells depleted for Seb1 showed reduced Rpa2-myc ChIP signal along the rDNA transcribed regions (Fig. 2b, beige versus gray bars) despite similar Rpa2-myc expression in wild-type and $P_{nmt1}$-seb1 cells (Fig. 2c). As a control, thiamine-dependent depletion of an unrelated essential protein did not affect Rpa2-myc occupancy at the rDNA (Supplementary Fig. 2b), indicating that the reduced RNAPI occupancy detected after Seb1 depletion is not the indirect consequence of reduced growth rate. We also confirmed that rDNA copy number was not affected by a deficiency in Seb1 (Supplementary Fig. 2c). To confirm the Seb1-dependent decrease in RNAPI density at the rDNA using an independent Seb1 conditional system, we used anchor-away[40] that allows rapid rapamycin-dependent nuclear exclusion of Seb1 (Supplementary Fig. 2d). Consistent with our data using the $P_{nmt1}$-seb1 conditional strain, rapamycin-dependent nuclear depletion of Seb1 for 2 h resulted in decreased RNAPI density compared to rapamycin-treated wild-type cells (Supplementary Fig. 2e and g).

To assess whether the reduced RNAPI occupancy detected in Seb1-depleted cells could be the consequence of impaired transcription initiation, we analyzed the binding of Rrn3, a transcription factor that is required for recruitment of RNAPI to the rDNA promoter[41,42]. As expected, ChIP analysis of Rrn3-myc in wild-type cells demonstrated strongest binding at the rDNA promoter (Fig. 2d, region B) and background signal inside the rRNA gene body (Fig. 2d, region 2). In contrast to Rpa2 levels, which generally decreased by >3-fold along the rDNA transcribed regions in Seb1-depleted cells (Fig. 2b), levels of Rrn3-myc at the rDNA promoter were not significantly reduced after thiamine-dependent Seb1 depletion (Fig. 2d, e). Together, these data suggest a role for Seb1 in rRNA transcription at a step beyond RNAPI initiation.

## Seb1 promotes processing of the 35 S rRNA primary transcript

To assess whether the reduced RNAPI density observed in Seb1-depleted cells affected rRNA production and processing, we examined the steady levels of rRNA precursors in wild-type and $P_{nmt1}$-seb1 conditional strains by northern blotting. Using a probe complementary to sequences in the ITS1 region (see Supplementary Fig. 3a), we were able to detect the uncleaved 35 S primary rRNA transcript (Fig. 3a, lane 1), the 33 S and 32 S pre-rRNAs resulting from cleavages at $A_0$ and $A_1$, respectively, in the 5'-ETS region (Supplementary Fig. 3a), and the $27SA_2$ that results from an endonucleolytic cleavage at site $A_2$ (Supplementary Fig. 3a). Interestingly, the depletion of Seb1 caused a marked accumulation of the 35 S, 33 S, and 32 S rRNA precursors together with reduced levels of $27SA_2$ pre-rRNA (Fig. 3a, compare lane 1−2). Quantification from independent experiments revealed a 4-fold increase in the $35 S/27SA_2$ ratio in Seb1-deficient cells (Fig. 3b). A significant increase in the $35 S/27SA_2$ ratio was also observed after nuclear depletion of Seb1 using the rapamycin-dependent anchor away system (Supplementary Fig. 3b, c). Analysis of short RNAs produced from 5'ETS processing revealed reduced cleavage at $A_0$ and $A_1$ in the seb1 mutant (Supplementary Fig. 3d, e), consistent with the accumulation of 35 S, 33 S, and 32 S pre-rRNAs (Fig. 3a).

We next analyzed pre-rRNA processing kinetics by pulse-chase labeling assays. Specifically, cells were pulsed with $[5,6-^3H]$-uridine to label the RNA backbone cotranscriptionally and then chased with excess unlabeled uridine, with samples harvested at various times points. In addition to the 35 S primary transcript, multiple pre-rRNA species were detected after pulse labeling (Fig. 3c, lane 1), reflecting both cotranscriptional and posttranscriptional pre-rRNA processing. After chasing for 3, 6, 12, and 45 min with excess of unlabeled uridine in wild-type cells, the 35 S primary transcript was cleaved into 27 S and 20 S pre-rRNA species, which were then processed into mature 25 S and 18 S rRNAs, respectively (Fig. 3c, lanes 2−5). Notably, Seb1-deficient cells showed delayed kinetics in processing the 35 S pre-rRNA into 27 S and 20 S precursors, resulting in reduced detection of mature 25 S and 18 S rRNA after 45 min (Fig. 3c, lanes 7−10). The significant delay in 35 S processing in Seb1-depleted cells (Fig. 3d) is consistent with the accumulation of 35 S primary transcripts detected by Northern blot assays (Fig. 3a). Together, our RNA analyses suggest that the cotranscriptional recruitment of Seb1 to the rRNA gene is important for pre-rRNA processing, most notably for the cleavage of the 35 S primary transcript at site $A_2$.

## The RNA binding function of Seb1, but not its RNAPII CTD-interactions, is required for rRNA processing

Fission yeast Seb1 is an evolutionarily conserved protein with two key functional domains: an Rpb1 CTD-interaction domain (CID) and an RNA recognition motif (RRM). We and others have previously used CID and RRM mutants to show that both domains of Seb1 are important for poly(A) site selection and transcription termination at RNAPII-transcribed genes[19,20]. We therefore used these previously characterized mutant versions of Seb1 to assess the functional significance of the Seb1 CID and RRM domains in RNAPI transcription and pre-rRNA processing. As shown in Fig. 4a, we used Seb1 versions with single and double substitutions at conserved K121 and K124 residues to glutamic acid (E) that impair binding to Ser2-phosphorylated versions of the CTD[20] as well as a S22A/K25A double mutant that affects binding to Ser5-phosphorylated CTD[19,20]. In the RRM, we used a mutant in which R442, H443, F445, and K447 residues were substituted to alanine (Fig. 4a). This $RRM_{MUT}$ affects poly(A) site selection and transcription termination at RNAPII-transcribed genes[19]. Wild-type and mutant alleles of seb1 were chromosomally integrated as a single copy in the $P_{nmt1}$-seb1 conditional strain (Fig. 4a), and the extent to which the Seb1 mutants restored the rRNA processing defects induced by the depletion of endogenous Seb1 was examined by northern blot assays. Western blot analysis confirmed that wild-type and mutant versions of Seb1-Flag were expressed at similar levels after thiamine-dependent depletion of endogenous Seb1 (Supplementary Fig. 4). As a positive control, expression of the wild-type version of Seb1 in the $P_{nmt1}$-seb1

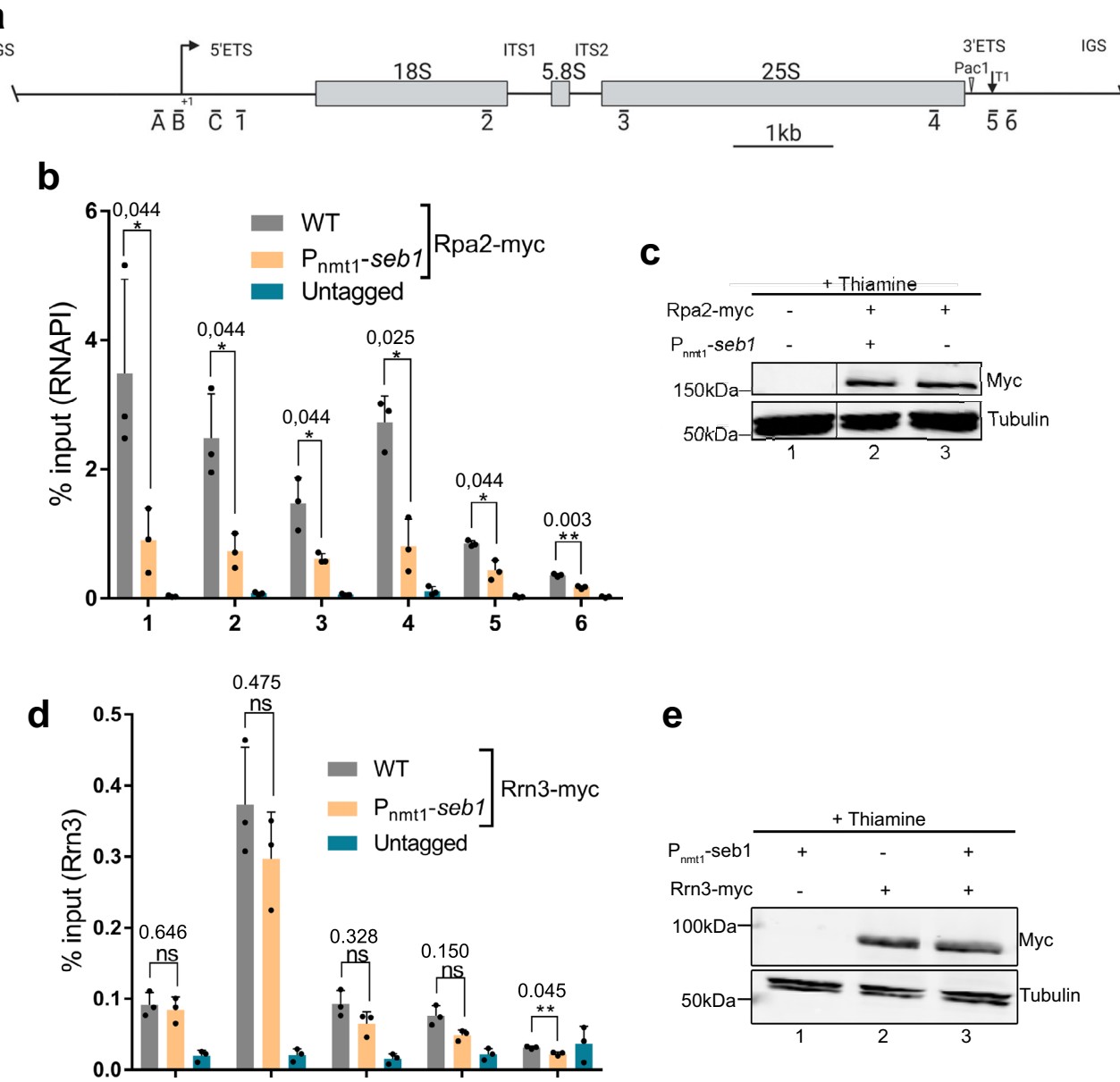

**Fig. 2 | Seb1 deficiency results in reduced RNAPI density at the rDNA.**
**a** Schematic of the rRNA gene. The bars under the rDNA indicate the positions of the PCR products used in the ChIP-qPCR analysis. **b** ChIP-qPCR analysis of RNAPI subunit (Rpa2-myc) in wild-type and *Pnmt1-seb1* strains on the rDNA repeats after the addition of thiamine for 10–12 h. Data and error bars represent the mean and standard deviation of $N = 3$ independent experiments. *$P$-value < 0.05; **$P$-value < 0.01, as determined by unpaired two-tailed Student's *t*-test corrected for multiple comparisons using the Holm-Sidak method. **c** Western blot analysis of Rpa2-myc in wild-type (lane 3) and *Pnmt1-seb1* (lane 2) strains following the addition of thiamine. The data in lanes 1–3 were from the same blot, with the vertical line indicating some

intervening lanes that were cropped out. The experiment was done $N = 2$ from independent biological replicates. **d** ChIP-qPCR analysis of Rrn3-myc in wild-type and *Pnmt1-seb1* strains on the rDNA repeats after the addition of thiamine for 10–12 h. Data and error bars represent the mean and standard deviation of $N = 3$ independent experiments. ns, $P$-value > 0.05; **$P$-value < 0.01, as determined by unpaired two-tailed Student's *t*-test corrected for multiple comparisons using the Holm-Sidak method. **e** Western blot analysis of Rrn3-myc in wild-type (lane 2) and *Pnmt1-seb1* (lane3) strains following the addition of thiamine. The experiment was done $N = 2$ from independent biological replicates.

strain prevented the thiamine-dependent increase in the 35 S/27 S ratio compared to the empty vector (Fig. 4b, compare lane 3 to lane 2; quantifications shown in Fig. 4c). Notably, all Seb1 mutants that impair interaction with the RNAPII CTD were able to fully restore the rRNA processing defects (Fig. 4b, compare lanes 5-8 to lane 2). In contrast, the version of Seb1 with substitutions that affect its RNA-binding function showed a significant increase in the 35 S/27 S ratio (Fig. 4b, compare lanes 4 to lane 3; Fig. 4c). These results indicate that the RNA-binding function of Seb1, but not its association with the RNAPII CTD, is required for Seb1-dependent rRNA processing.

## Seb1 controls RNAPI progression
Our ChIP assays indicate that the absence of Seb1 results in reduced RNAPI density at the rDNA (Fig. 2 and Supplementary Fig. 2). Moreover, Seb1-deficient cells showed delays in cleavage of the 35 S primary transcript (Fig. 3 and Supplementary Fig. 3), a process that is primarily cotranscriptional in yeast[27]. These findings could be explained by a faster transcription elongation rate, which is known to modulate cotranscriptional RNA processing[13,43]. To test whether Seb1 influences the elongation kinetics of RNAPI in vivo, we determined the distribution of RNAPI at the single nucleotide resolution by CRAC, a

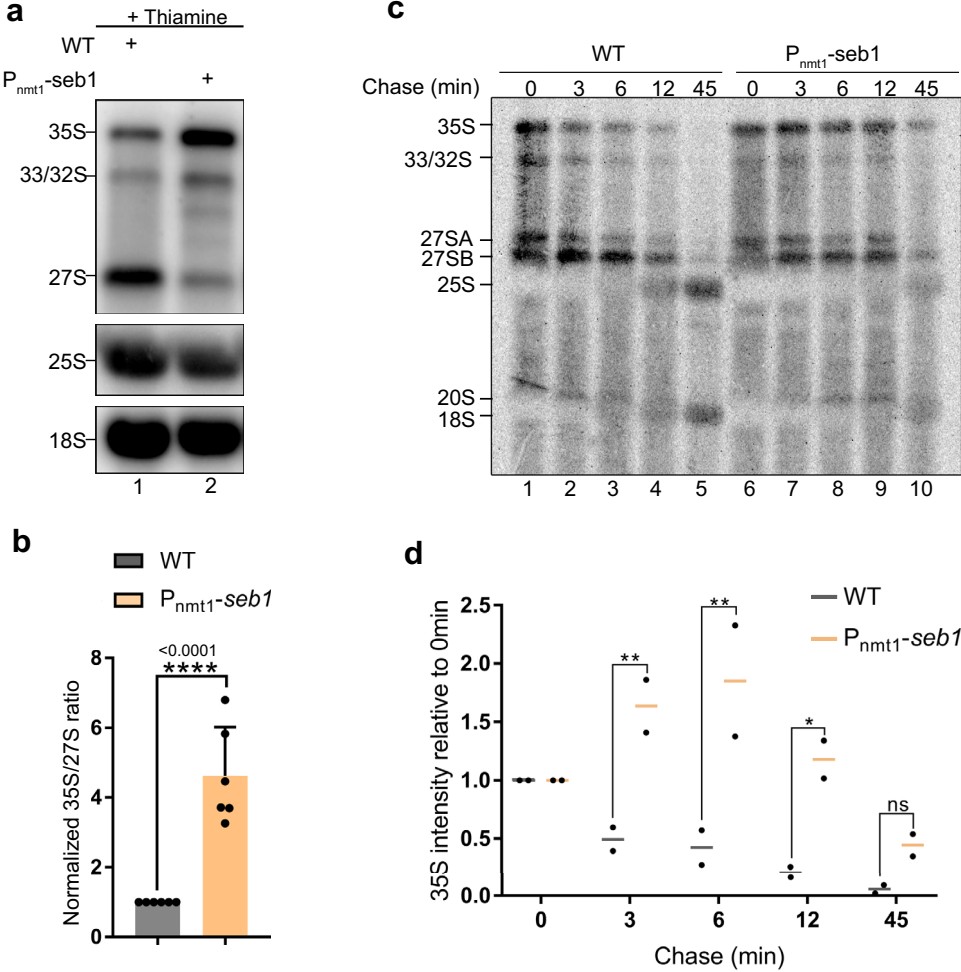

**Fig. 3 | Seb1 is required for efficient pre-rRNA processing. a** Northern Blot analysis of total RNA prepared from wild-type (lane 1) and *Pnmt1-seb1* (lane 2) strains following addition of thiamine for 10–12 h. *Top*, the membrane was analyzed using a probe complementary to ITS1-specific sequences (see Supplementary Fig. 3a). *Middle and bottom*, mature rRNAs were analyzed using probes complementary to 25S- and 18S-specific sequences, respectively. Pre-rRNAs and mature rRNAs are indicated on the left. The experiment was done $N = 6$ from independent biological replicates. **b** Quantification of 35 S/27 S pre-rRNA ratio in the *Pnmt1-seb1* strain relative to the wild-type strain. Data and error bars represent the mean and standard deviation of $N = 6$ independent experiments. ****$P$-value < 0.0001, as determined by unpaired two-tailed Student's *t*-test. **c** Pulse-Chase analysis of total RNA prepared from wild-type (lanes 1–5) and *Pnmt1-seb1* (lanes 6–10) strains following the addition of thiamine for 10–12 h. The cells were then pulse-labeled with [5,6H]-uridine for 4 min and chased with an excess of unlabeled uridine. Total RNA was extracted from cells samples harvested at the indicated time points and resolved on a 0.8% agarose–formaldehyde gel. The position of the rRNA species is indicated on the right. The experiment was done $N = 2$ from independent biological replicates. **d** Levels of 35 S pre-rRNA in wild-type and *Pnmt1-seb1* cells relative to $t = 0$ min at the indicated time points. Datapoints from two biologically independent experiments are shown. ns, $P$-value > 0.05; *$P$-value < 0.05; **$P$-value < 0.01; as determined by Student's *t*-test.

high-resolution approach that was previously used to map the elongation profiles of budding and fission yeast RNAPI[36]. To perform CRAC analysis of RNAPI, we HTP-tagged *S. pombe* Rpa2 at its endogenous chromosomal locus in both wild-type and *Pnmt1-seb1* conditional strains and confirmed the reduction of Rpa2-HTP density at the rDNA in Seb1-deficient cells by ChIP assays (Supplementary Fig. 5a, b). Following growth in thiamine-supplemented medium to deplete Seb1, RNA-protein crosslinks were induced by treating cells using 254-nm UV irradiation. Next, tandem affinity purification of Rpa2-HTP from extracts of Seb1-depleted and control cells under stringent denaturing conditions enriched for RNAPI-associated nascent pre-rRNAs, followed by on-beads ligation of adapters and preparation of cDNA libraries for high-throughput sequencing. Note that a phosphatase treatment was omitted from the procedure to recover nascent transcripts with native 3'-OH groups. Accordingly, data analysis exclusively used the 3' end of reads, which is expected to represent the 3' end of the nascent pre-rRNA inside the catalytic core of the RNAPI complex. Regions with high signal (peaks) are interpreted as having high RNAPI occupancy and,

therefore, limited transcription progression and/or stalled transcription elongation complex. Conversely, valleys reflect low RNAPI occupancy and swift transcription progression. As expected, Rpa2-HTP showed predominant recovery of rDNA-transcribed species (Supplementary Fig. 5c). Consistent with previous findings[36], the overall CRAC profile of RNAPI distribution along the *S. pombe* rDNA showed a strong enrichment in the 5'-ETS region (Fig. 5a), reflecting slower RNAPI progression and/or more frequent pausing in the 5'ETS region of the rDNA. Comparison of CRAC data between wild-type and Seb1-depleted cells revealed reduction of the most prominent peaks in cells deficient for Seb1 (Fig. 5a, compare green and blue profiles), which is consistent with decreased RNAPI pausing time and/or more homogenous transcription progression compared to wild-type. The most affected areas were in the 5'-ETS (Fig. 5b, compare green and blue peaks) and ITS1-5.8S-ITS2 (Fig. 5c) regions of the rRNA gene. Notably, visualizing the difference in sequencing coverage in the ITS1-5.8S-ITS2 region showed 2- to 4-fold variations in RNAPI presence between wild-type and Seb1-deficient cells (Fig. 5c, *bottom panel*). Quantification of relative read

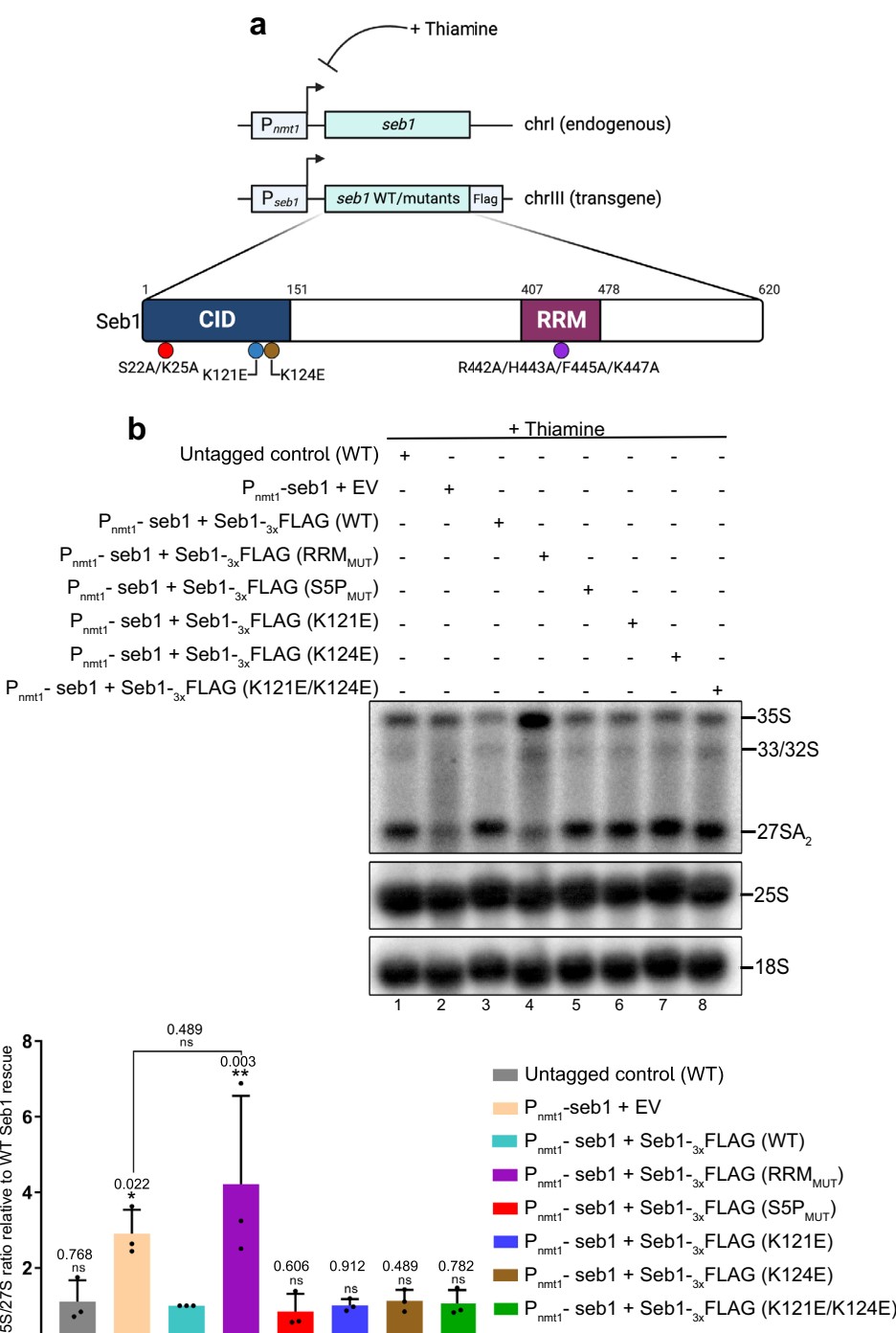

**Fig. 4 | Seb1-dependent pre-rRNA processing requires its RNA-binding function. a** Schematic of Seb1 complementation assay. See Result section for details. **b** Northern Blot analysis of pre-rRNA from the indicated strains after the addition of thiamine for 10–12 h. *Top*, the membrane was analyzed using a probe complementary to ITS1-specific sequences. *Middle and bottom*, mature rRNAs were analyzed using probes complementary to 25S- and 18S-specific sequences, respectively. Pre-rRNAs and mature rRNAs are indicated on the right.

**c** Quantification of 35 S/27 S pre-rRNA ratio in the indicated strains expressed relative to Seb1-depleted cells complemented with wild-type Seb1-Flag. EV, empty vector. Data and error bars represent the mean and standard deviation of $N = 3$ independent experiments. ns, *P*-value > 0.05; *P*-value < 0.05; **P*-value < 0.01; as determined by unpaired two-tailed Student's *t*-test corrected for multiple comparisons using the Holm-Sidak method.

counts also revealed substantially decreased pausing over the 5.8 S and ITS2 region (Fig. 5d). Further, by converting the fraction of reads into occupancy timescale (see methods), the absence of Seb1 showed considerable differences in average RNAPI transcription time in the ITS1-5.8S-ITS2 region, dropping to average occupancy intervals of 50 ms at specific positions where the occupancy time was >150 ms in wild-type cells (Supplementary Fig. 5d). Interestingly, despite CRAC

profiles showing less prominent RNAPI peaks along the rDNA when Seb1 is lacking (Fig. 5a–d), we found increased peak densities at the 3' end of the rRNA gene, just upstream of RNAPI transcription termination (Fig. 5e, note blue versus green peak around position +7715). We also noted a greater 5' peak at position +36 (Fig. 5b), which is thought to represent RNAPI in an initiation-related stalled/paused state[36,41]. Collectively, our CRAC data indicate that a Seb1 deficiency leads to

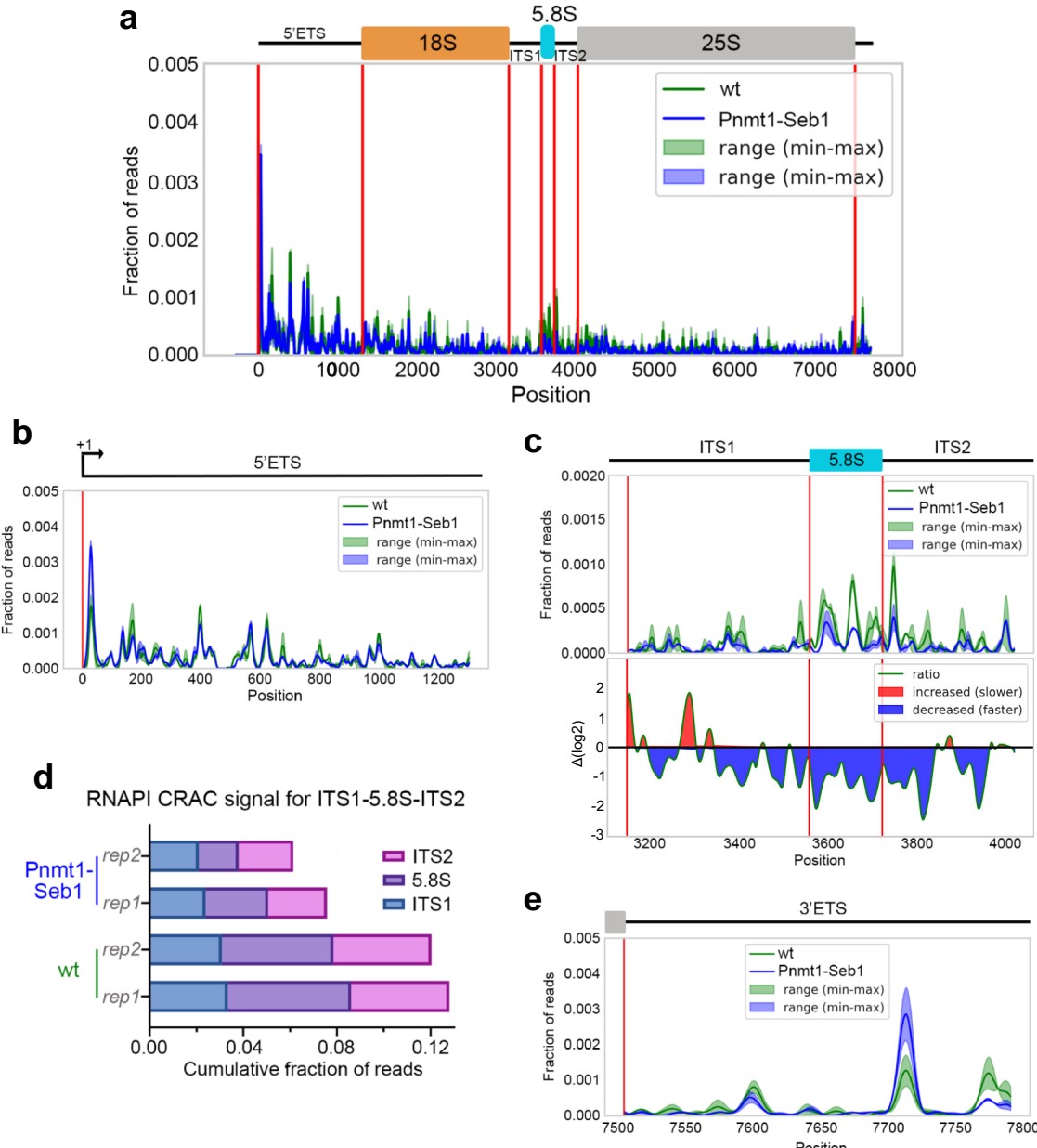

**Fig. 5 | Seb1 delays RNAPI progression along the rDNA. a** RNAPI CRAC distribution over the rRNA gene. *Top*, schematic representation of the pre-rRNA transcription unit, including 18 S, 5.8 S, and 25 S rRNA as well as ETS and ITS regions. *Bottom*, Rpa2 CRAC profiles, presented as fractions of reads, for wild-type (green) and *Pnmt1-seb1* (blue) cells after the addition of thiamine for 10–12 h. The solid dark lines mark the median for two biological replicates, while the pale profiles indicate the range between second and third quartile. **b** RNAPI CRAC profiles across the first 1300 nt of the transcription unit corresponding to the 5'ETS. **c** *Top*, RNAPI CRAC profiles across the ITS1, 5.8 S, and ITS2 regions of the transcription unit; *Bottom*, Log2-transformed ratios (Δlog2) between Seb1-depleted and wild-type control are displayed. Regions showing decrease and increase RNAPI occupancy in Seb1-depleted cells are marked with blue and red, respectively. **d** Comparison of the distribution of CRAC signal across the ITS1, 5.8 S, and ITS2 regions of the rRNA transcription unit for independent replicates (rep) of wild-type (wt) and Seb1-deficient (Seb1) cells. **e** RNAPI CRAC profiles across the 3'ETS region of the transcription unit.

decreased pausing/slow states during RNAPI transcription elongation, especially over the ITS1-5.8S-ITS2 region which is crucial for cotranscriptional rRNA processing.

## Discussion

Recent analysis of yeast RNAPI distribution at single nucleotide resolution have established that the rate of RNAPI elongation is highly dynamic, varying extensively within the rRNA gene[35,36,44]. To date, however, how the elongation rate of RNAPI transcription is controlled remains poorly understood. In this study, we show that the evolutionarily conserved protein Seb1 is required for optimal cotranscriptional

rRNA processing by promoting RNAPI pausing and/or slower elongation states. Given that Seb1 also promotes polymerase pausing to control poly(A) site selection and heterochromatin assembly during RNAPII transcription[19–21], we propose that Seb1 acts as a pause-promoting factor for RNA polymerases I and II to facilitate cotranscriptional RNA processing.

Our data indicated that a Seb1 deficiency results in defects in rRNA processing and RNAPI transcription, showing decreased pausing/slow states during RNAPI elongation. Because Seb1 affects mRNA 3' end processing and termination of RNAPII-transcribed genes[19,20], we cannot totally exclude the possibility that the effects of the depletion of

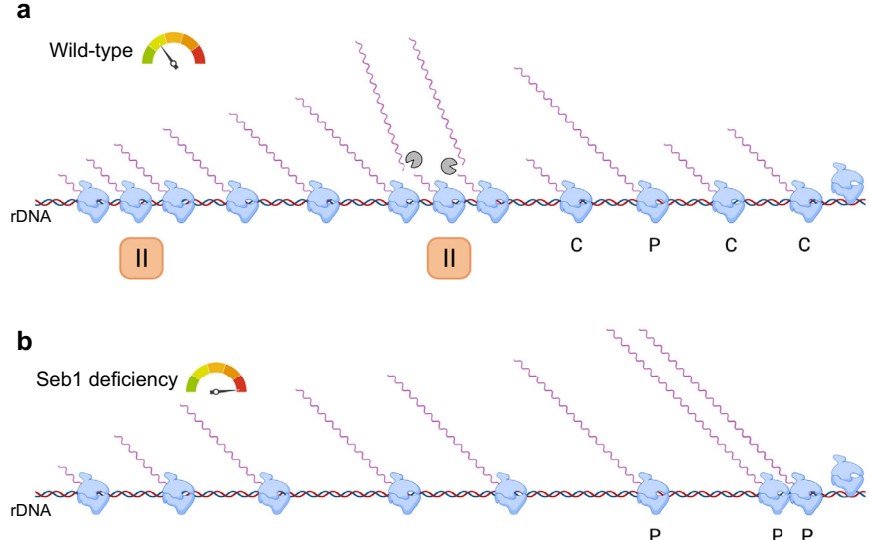

**Fig. 6 | Model for how Seb1 promotes cotranscriptional pre-rRNA processing.**
**a** NET-seq and CRAC analysis in wild-type budding and fission yeasts show that RNAPI frequently pauses when transcribing the long rRNA gene[35,36]. We propose that the cooperative binding of Seb1 to the RNAPI complex and to nascent pre-rRNA contributes to RNAPI pausing, thereby facilitating recognition of processing sites by the SSU-processome (gray pacman), which will cotranscriptionally (C) cleave the nascent pre-rRNA. A minority of pre-rRNAs that are not processed cotranscriptionally will be cleaved posttranscriptionally (P) after transcription termination[27]. **b** In the absence of Seb1, RNAPI progresses faster along the rRNA gene, impairing the efficiency of cotranscriptional pre-rRNA processing.

Seb1 on rRNA processing and RNAPI transcription are indirect. Yet, given that Seb1 physically associated with the RNAPI transcription machinery, was recruited to rDNA (whereas other mRNA processing factors Rna14, Pcf11, Ysh1, and Cbp80 were not), and was shown to crosslink to pre-rRNA, the simplest model is that loss of Seb1 directly influences RNAPI progression, leading to defective cotranscriptional rRNA processing. The fact that Seb1 mutants defective in binding phosphorylated versions of the RNAPII CTD did not impair rRNA processing (Fig. 4) also argues for a direct role of Seb1 on RNAPI transcription. In support to the conclusion that Seb1 contributes to both RNAPI and RNAPII transcription is evidence that other factors that influence the elongation phase of RNAPII transcription, such as the Spt4/Spt5 and Paf1 complexes, have also been demonstrated to contribute to transcription elongation of RNAPI[34,45,46].

Our ChIP analyses of RNAPI at the rDNA revealed decreased polymerase density in the absence of Seb1 (Fig. 2). Polymerase density along a gene is determined by three factors: (1) initiation frequency at the promoter, (2) the extent of premature termination, and (3) the speed with which polymerases elongate the nascent RNA through the various sections of a gene. The similar enrichment levels of Rrn3, which is required for the recruitment of RNAPI to the rDNA promoter[41,42], in wild-type and Seb1-deficient cells argues against defective RNAPI initiation. In fact, a lower RNAPI initiation frequency per se is not expected to result in pre-rRNA processing defects, which was the case in the absence of Seb1. The extent of premature termination during RNAPI elongation is poorly understood, but has been recently suggested as a mechanism that could control rRNA production[47,48]. Although we cannot entirely rule out that Seb1 contributes to RNAPI processivity (prevents premature termination), our ChIP assays of Rpa2 did not appear to show a biased decrease at the 3' end of the rRNA gene relative to its 5' (Supplementary Fig. 2f, g), which is expected if RNAPI processivity is impaired[45]. Importantly, our analysis of RNAPI distribution at single nucleotide resolution by CRAC demonstrated that a Seb1 deficiency results in decreased pausing/slow states during RNAPI transcription. Given that formaldehyde crosslinking during ChIP requires a minimal chromatin residence time[49], a fast transcribing polymerase is expected to show reduced crosslinking efficiency. Thus, the simplest interpretation of the Seb1-dependent reduction in RNAPI density as determined by ChIP is faster transcription progression on the rRNA gene in Seb1-deficient cells.

It is well known that the rate of transcription elongation by RNAPII directly impacts pre-mRNA processing decisions such as splicing and 3' end processing[50]. This is also true for RNAPI, as a single mutation in a gene that codes for a subunit of *S. cerevisiae* RNAPI that impairs transcription elongation results in rRNA processing defects[33,44]. Coupling between RNAPI transcription and rRNA processing is further supported by the fact that roughly 70% of the endonucleolytic cleavage at site $A_2$ in the ITS1 region, which generates the 27 S rRNA precursor, occurs cotranscriptionally on the nascent primary transcript[27]. In the current study, we found that loss of Seb1 results in pre-rRNA processing defects that delay the production of mature rRNAs (Fig. 3). Specifically, our results indicated that the cleavage efficiency at sites $A_0$, $A_1$, and $A_2$ were significantly decreased in Seb1-deficient cells, leading to an increase in the 35 S/27 S ratio. Given the overall faster progression of RNAPI on the rRNA gene in the absence of Seb1, especially in the ITS1-5.8S-ITS2 region (Fig. 5c, d) where Seb1 binds directly (Fig. 1h) and cleavage at site $A_2$ occurs, we suggest that the Seb1-dependent changes in RNAPI progression along the rDNA impair the efficiency of cotranscriptional rRNA processing (Fig. 6).

Analysis of Seb1-RNAPII associations by crosslinking-mass spectrometry reveals abundant contacts with regions near the RNA exit channel of RNAPII[18]. Given the overall structural similarity between the catalytic core of RNAPI and RNAPII[22,23], we suspect that similar protein-protein interactions between Seb1 and the RNAPI complex (Fig. 1d–f) contribute to the recruitment of Seb1 to rDNA during RNAPI transcription, as demonstrated by our ChIP assays (Fig. 1a–c). Seb1-RNA interactions with the nascent pre-rRNA (Fig. 1h) at the site of RNA exit of the transcribing polymerase also likely contribute to how Seb1 influences cotranscriptional RNA processing. Consistent with the idea that RNA binding by Seb1 is important to promote cotranscriptional rRNA processing, our results revealed that a functional RRM domain in Seb1 was required for pre-rRNA processing (Fig. 4) and recruitment to the rDNA locus (Supplementary Fig. 1b). This conclusion echoes findings on the role of Seb1 in RNAPII-derived transcription, indicating that the RNA binding function of Seb1 is required for Seb1-dependent poly(A) site selection[19] and heterochromatin assembly[21].

Seb1 controls the progression of RNAPI (this study) and RNAPII[21]. Although the detailed mechanism by which Seb1 influences transcription elongation kinetics remains to be determined, it is tentative to speculate about opposing roles of Seb1 and the Spt4/Spt5 complex. First, Seb1 appears to contact sites on RNAPII that overlap with surface areas bound by Spt5 as well as compete with Spt5 for RNAPII binding in vitro[18]. Second, the Spt4/Spt5 complex was shown to function as a positive regulator of transcription elongation by RNAPI[45,51] and RNAPII[52,53], whereas Seb1 appears to promote pausing of RNAPI (this study) and RNAPII[21]. It is also possible that Seb1 promotes polymerase backtracking, which is a major mechanism underlying pausing of RNA polymerases[8]. During backtracking, newly transcribed RNAs outside of the polymerase exit channel must re-enter the polymerase catalytic center as the enzyme is slipping backward. The presence of the RNA-binding domain of Seb1 close to the RNA exit channel of the polymerase may prevent the untimely formation of RNA secondary structures, which have been shown to refrain backtracking[36,54,55]. Future studies using in vitro transcription systems should provide insights into the underlying mechanism by which Seb1 promotes long-live RNAP pausing.

Whereas Seb1 is important for termination of RNAPII transcription[19,20], our ChIP analysis did not reveal defects in RNAPI termination in Seb1-deficient cells. This was somewhat surprising, as transcription termination of budding yeast RNAPI relies on cotranscriptional RNA cleavage by the dsRNA endoribonuclease Rnt1, which provides entry for the 5'–3' exonuclease Rat1 that degrades RNAPI-associated transcripts and destabilizes the RNAPI transcription complex[37,38]. The faster RNAPI progression in Seb1-depleted cells may be expected to impair such cotranscriptional RNA cleavage in the 3'ETS of the rRNA gene, leading to read-through RNAPI. However, we have recently reported that the fission yeast homolog of *S. cerevisiae* Rnt1, Pac1, is not required for RNAPI termination in *S. pombe*[56], suggesting that fission yeast relies on redundant/alternative mechanisms for termination of RNAPI transcription. One possibility is the "road-block" mechanism that was previously described in budding yeast[57,58], which involves a DNA-binding factor that promotes transcription termination by pausing/blocking incoming RNAPI at the 3' end of the rDNA. Consistent with this view, our CRAC data revealed piled-up RNAPI at the rDNA 3' end in Seb1-deficient cells (Fig. 5e), which may be expected in a model whereby RNAPI complexes traverses the rRNA gene body with reduced interruptions (Fig. 6b).

Proliferating yeast and human cells produce around 2000 and 7500 ribosomes per minute[25,59], a process that heavily relies on efficient pre-rRNA processing. The identification of Seb1 as an essential factor that promotes cotranscriptional pre-rRNA processing is an important advance in understanding the interplay between rRNA transcription and processing. Given that fission yeast Seb1 promotes cotranscriptional RNA processing by controlling progression of RNAPI (this study) and RNAPII[19,21], together with the fact that human Seb1 homologs, SCAF4 and SCAF8, also control cotranscriptional mRNA processing by RNAPII[60], we predict that the link between transcription elongation kinetics and pre-rRNA processing described in yeast are likely to apply to metazoans.

## Methods

### Yeast strain and media
A list of *S. pombe* strains and oligonucleotides used in this study can be found in Supplementary Table 1 and 2. Unless otherwise specified, cells were grown at 30 °C to their mid-log exponential phase ($OD_{600nm}$ ~ 0.5–0.8) either in YES (Yeast extract medium) or EMM (Edinburg minimal media) supplemented with adenine, histidine, uracil and leucine. For the cells where the *seb1* gene was under the thiamine-sensitive *nmt1* promoter, treatment with 60 μM of thiamine for 10-14 h were performed to repress transcription. Nuclear depletion of Seb1 was performed using the anchor-away system[40] after addition of 2.5 μg/mL rapamycin for 2 h. Mutations in Seb1 were induced by site-directed mutagenesis using primers containing the induced mutations, as previously described[19]. Addition of C- and N-terminal molecular tags and deletion of genes were performed by a PCR-mediated gene targeting approach[61]. Expression of proteins with epitope tags were confirmed by western blots and gene deletion was validated by RT-qPCR.

### Microscopy
Seb1-FRB-GFP localization was detected using fluorescence microscopy as previously described[7]. Briefly, liquid cultures were grown in EMM to early log phase ($OD_{600nm}$ ~ 0.3) then rapamycin or an equal volume of DMSO was added to a final concentration of 2.5 μg/ml. After 2 h incubation, nuclei were stained using Hoechst 33342 for 15 min (0.2 mg/ml) and live cells were mounted on 1.2% agarose patches. GFP-tagged proteins and nuclei were detected at 470 nm and 365 nm, respectively, using a Colibri system (Carl Zeiss Canada, Toronto, ON, Canada) on a Zeiss Axio Observer Z1 inverted microscope with a ×100/1.4 oil objective. Data were analyzed using the ZEN black software (Carl Zeiss Canada).

### Cross linking and cDNA analysis (CRAC)
CRAC analysis of Rpa2-HTP was performed as previously described[36] from two independent biological replicates. Briefly, strains for CRAC analysis were grown in EMM medium supplemented with thiamine to allow Seb1 depletion. Mid-log exponential phase ($OD_{600nm}$ ~ 0.5–0.8) cells were cross-linked in culture media using megatron UVC cross-linker[36] for 100 s. Crosslinked samples were processed and lyzed in TNMC100 (50 mM Tris-HCl pH 7.5, 150 mM NaCl, 0.1% NP-40, 5 mM $MgCl_2$, 10 mM $CaCl_2$, 5 mM β-mercaptoethanol, 50U of DNase RQ1 and a protease-inhibitor cocktail) with zirconia beads in a 50 mL conical. Cleared lysates were incubated with IgG Sepharose and washed washed three times with TMN600 (50 mM Tris-HCl pH 7.5, 600 mM NaCl, 0.1% NP-40, 1.5 mM $MgCl_2$) and two times TMN100 (50 mM Tris-HCl pH 7.5, 100 mM NaCl, 0.1% NP-40, 5 mM $MgCl_2$). Protein:RNA complexes were eluted with HaloTEV and the supernatant was subjected to nickel affinity purification in denaturing conditions. After extensive washing of the nickel beads, the addition of 3' linkers, 5' end radiolabeling, and 5' linker ligation was performed on beads. Protein:RNA complexes were then eluted, precipitated, separated on gels, and transferred to nitrocellulose membranes. After detection of labeled RNAs by autoradiography, the appropriate region was excised and treated with proteinase K. The RNA was then isolated, reverse transcribed and PCR amplified as previously described[36]. Reverse transcription and PCR amplification of wild-type and Seb1-depleted samples were analyzed simultaneously to minimize PCR amplification bias during library preparation. Following gel extraction of PCR amplicons corresponding to 140-200 bp, libraries were measured by Qubit and analyzed by Illumina sequencing.

The CRAC data was analysed as previously described[36]. Downstream analysis and visualization were performed in Python using pandas, matplotlib and trxtools (https://github.com/TurowskiLab/trxtools). In short, the 3' ends of the reads were counted and pseudocounts were added. The 3' ends of the reads were then transformed into fraction of reads for the transcription unit, smoothed with 30-nt blackman window (pandas) and two biological replicates were averaged. To calculate ratio between datasets, the data were smoothed with 50-nt window, the Seb1-depleted profile was divided by the wild-type profile, and log2 transformation was applied. To calculate the occupancy time (Supplementary Fig. 5d), the total transcription (occupancy) time $T_{total}$ was calculated by dividing the length of the rDNA transcription unit (7800 nt) by the average velocity of yeast RNAPI[27] (40 nt·s⁻¹). Occupancy time for each position was calculated by multiplication of fraction of reads for each position with $T_{total}$.

## Protein immunoblotting

Total cell extracts were resuspended in a cold lysis solution (50 mM Tris [pH 7.5], 5 mM MgCl2, 150 mM NaCl and 0.1% NP-40) supplemented with protease inhibitors (1X PMSF and 1X PLAAC). Following the addition of glass beads, lysis on a fast prep (MP Biomedical) was performed with 3 cycles of 30 s at 6.5 m/s. Final protein concentrations were measured and 30 µg of total proteins were separated on an SDS-PAGE gel, transferred to a nitrocellulose membrane, and analyzed by immunoblotting using either a (i) mouse monoclonal anti-myc (9E10, sc-40, 1:500 (v/v) dilution) or anti-Flag (Sigma-Aldrich, F1804; 1:500 (v/v) dilution) or a (ii) rabbit polyclonal antibody against protein A (Sigma-Aldrich, P3775; 1:10,000 (v/v) dilution) in the case of HTP-tagged Seb1. Membranes were probes with goat anti-mouse secondary antibodies coupled to AlexaFluor 680 (Life Technologies, A-21057; 1:15,000 (v/v) dilution) Protein detection was performed with the Odyssey infrared imaging system at 680 nm (LI-COR).

## Chromatin immunoprecipitation (ChIP) assays

ChIP assays were performed as previously described[7,16,19]. For each condition (IP), 50 mL of cell cultures were grown to an $OD_{600nm}$ of ~0.5–0.8 at 30 °C in EMM + 60 µM of thiamine for 12-15 h. Then, cross-linking was performed by adding a final concentration of 1% formaldehyde in a diluent solution (0.143 mM NaCl, 1.43 mM EDTA and 71.33 mM HEPES-KOH [pH7.5]) for 20 min at room temperature (RT) with manual shaking every 5 min. After quenching the solution with glycine (final concentration of 360 mM), cells were washed twice with a cold Tris-buffer saline (20 mM Tris-HCl [pH 7.5] and 150 mM NaCl) and frozen in liquid nitrogen. Cell pellets were thawed and resuspended in 500 µL of cold lysis buffer (50 mM HEPES-KOH [pH 7.5], 140 mM NaCl, 1 mM EDTA pH 8.0, 1% Triton X-100 and 0.1% Na-deoxycholate) that was supplemented with a protease inhibitor cocktail (1X PMSF and 1X PLAAC) and disrupted vigorously with glass beads in a fast prep (Mp Biomedical) for 3 cycles of 30 s at 6.5 m/s. Then, 150 µL of cold lysis buffer with proteases was added to increase the volume and samples were sonicated for 12 cycles of 10 s at 20% amplitude with a Branson digital sonifier. Whole cell extracts (WCE, 500 µl) were then incubated overnight at 4 °C with 50 µl of IgG Dynabeads (Life Technologies, 11041) in the case of HTP/TAP tagged proteins, or beads coupled with 1.5 ug of mouse monoclonal anti-myc (9E10, sc-40) or 2ug of anti-flag (Sigma-aldrich, F1804). Then, beads were washed twice with 1 mL of cold lysis buffer, twice with 1 mL of cold lysis buffer with 500 mM NaCl, twice with 1 mL of cold wash buffer (10 mM Tris-HCl [pH 8.0], 250 mM LiCl, 0.5% NP-40, 0.5% sodium deoxycholate and 1 mM EDTA) and once with 1 mL of cold Tris-EDTA (TE: 10 mM Tris-HCl [pH 8.0] and 1 mM EDTA [pH8.0]). Bound material was eluted by resuspending the beads in 50 µl elution buffer (50 mM Tris-HCl [pH 8.0], 10 mM EDTA and 1% SDS) at 65 °C for 15 min at 1200 rpm in an Eppendorf Thermomixer. After a short centrifugation, reverse-crosslinking was induced by incubating eluted material with 120 µl of TE and 1% SDS or 5 µl of WCE in 95 µl of TE with 1% SDS at 65 °C overnight. Samples were treated with a proteinase K mix (150 mg proteinase K, 60 mg glycogen in TE) and DNA was extracted twice with phenol-chloroform-isoamyl alcohol (25:24:1, [pH8.0], Invitrogen) and once with chloroform (Bioshop), then precipitated with ethanol and resuspended in 30 µl of TE. 10 µg of RNAse A was added to the DNA and after an 1 h incubation at 37 °C, the DNA was purified with a QIAGEN PCR purification kit. The inputs DNA (WCE) was diluted 100 times while the conditions DNA (IP) was diluted 20 times in water. DNA were analyzed on a LightCycler 96 Instrument system (Roche) with the addition of the supermix prerfecta SYBR (QuantaBio) in the presence of 150 mM of gene-specific oligonucleotide. Protein density was calculated by subtracting the mean of three technical replicates of CT conditions (IP) from CT input (WCE). Then, the difference was used to calculate the relative abundance of the immunoprecipitated protein using the $2\Delta CT$ formula.

## RNA preparation and analysis

Total RNA was extracted by the acid-phenol heating method, as described previously[56]. The cells are first resuspended in 500 µl of TES solution (10 mM Tris-HCl [pH 7.5], 10 mM EDTA [pH8.0] and 0.5% SDS) and 500 µl of phenol-chloroform 5:1 (Bioshop) then incubated at 65 °C for 60 min with vortexing every 10 min. The RNA was extracted once with 500 µl of phenol-chloroform 5:1 (Bioshop) and once with 500 µl of chloroform-isoamyl alcohol (24:1, Bioshop) then precipitated with ethanol and 3 M NaOAc [pH5.2]. For RT-qPCR analyses, 1 µg of total RNA was treated with DNASe RQ1 (Promega, M6101) and reverse transcribed with RT Omniscript (QIAGEN). The DNA was then diluted 100 times and the expression of the genes was measured on a LightCycler 96 Instrument system (Roche) with the addition of the supermix prerfecta SYBR (QuantaBio) in the presence of 150 mM of gene-specific oligonucleotide (Supplementary Table 2). The ΔΔCq method was used to calculate RNA abundance using the *nda2* gene (code for alpha 1 tubulin) as an internal reference gene. RNA abundance was plotted by normalizing the abundance of the mutant to the abundance of the *nda2* gene and either the abundance of a wild-type strain or a specific region.

Following total RNA extraction, 5 µg/well of RNA mixed with a volume of 0.66 of loading buffer were separated on a 0.8% agarose gel containing formaldehyde for 4h30 at 200 V. RNAs were transferred onto a nylon membrane (Amersham Hybond-XL, GE Healthcare) pre-hybridized in a 5X Denhardt buffer at 42 °C and crosslinked twice. DNA probes were 5' radiolabeled with [γ-$^{32}$P]-ATP using T4 polynucleotide kinase (NEB) for 30 min and heated at 95 °C for 5 min before being added to the membrane. The next day, membranes were washed twice with 2× SSC/0.1% SDS for 5 min and twice with 0.1× SSC/0.1% SDS for 15. A storage phosphor screen (GE healthcare) was added to the membrane and photos were taken at different times with a typhoon trio. Quantification was performed with ImageQuantTL software (GE healthcare).

## Pulse chase analysis of pre-rRNA

Pre-rRNA pulse chase assays were performed as previously described[62], with minor modifications. Briefly, 50 mL of cells were grown at 30 °C in EMM-ura to $OD_{600nm}$ of ~0.5−0.8. The cells were then diluted to an $OD_{600nm}$ of 0.018 (WT) or 0.03 (P*nmt1-seb1*) where 60 µM of thiamine was added for 15 h at 30 °C. Cells were resuspended in 850 µl of EMM-ura containing 125µCi of uridine-$^3$H (Perkin Elmer, cat no: NET367250UC). Following a 4 min pulse, 200 µl of cells were diluted in 1.7 mL of EMM supplemented with uracil and 60 µM of thiamine. At various times point, cells were taken and rapidly frozen in liquid nitrogen. RNA was purified and separated as described in the 'preparation and analysis of RNA' section, with the following modifications: 10,000 cpm of radioactivity from each sample were loaded and migrated onto a 0.8% agarose-formaldehyde gel. Finally, a tritium screen (BAS-IP TR 2025 E Tritium Screen, 28956482, Cytiva) was added to the membrane and analyzed suing a typhoon trio instrument.

## TurboID

TurboID assay analysis of Seb1 ($n = 1$) was performed as described previously[39]. 50 mL of cultures were grown in YES medium supplemented with 50 µM biotin to an $OD_{600nm}$ of ~0.5−0.6. Cell pellets were resuspended in cold 1X RIPA buffer (50 mM Tris-HCl [pH 7.5], 150 mM NaCl, 1.5 mM MgCl2, 1 mM EGTA, 0.1% SDS and 1% NP-40) supplemented with 0.4% sodium deoxycholate, 1 mM DTT, 1 mM PMSF, 1× PLAAC and 1x cOmplete, then disrupted vigorously with a Fast prep (Mp Biomedical) with 3 cycles of 30 s at 6.5 m/s. Sample volume was increased to 500 µl using the same buffer before sonication for three cycles of 10 s at 20% intensity using a Branson Sonifier 250. Then, 1 µl of benzonase 250U/µl (Sigma-Aldrich; E1014) was added at 4 °C for 1 h for DNA and RNA digestion. The whole cell extract (WCE) was then normalized for total protein concentrations by performing a Bradford protein assay. Then, 1 mL of cold 1X RIPA buffer with supplement and 0.4% total SDS (instead of 0.1%) containing 5 mg of protein was added

to 50 µl of Streptavidin–Sepharose beads (GE Healthcare; 17-5113-01) for 3 h at 4 °C. Beads were washed once with washing buffer (50 mM Tris-HCl [pH 7.5] and 2% SDS), three times with cold 1X RIPA buffer with addition of 1 mM DTT (50 mM Tris-HCl [pH 7.5], 150 mM NaCl, 1.5 mM MgCl2, 1 mM EGTA, 0.1% SDS and 1% NP-40) and five times with ABC buffer (20 mM ammonium bicarbonate). All washes were done for 5 min at room temperature with agitation. The proteins attached to the beads were then reduced by the addition of 10 mM DTT and alkylated by the addition of 15 mM iodoacetamide (IAA). To digest the proteins still bound to the beads, 1 µg of trypsin was added and incubated at 37 °C overnight and stopped by adding formic acid (final concentration of 1%). Peptides were then extracted twice with acetonitrile, lyophilized and resuspended in 0.1% trifluoroacetic acid (TFA). Desalting was done using ZipTips (EMD Milipore). After a second lyophilization, the peptides were resuspended in 25 µl of TFA then quantified with a nanodrop (Thermo Fisher) at 205 nm.

### LC-MS/MS analysis

Trypsin-digested peptides were separated using a Dionex Ultimate 3000 nanoHPLC system. Ten microliters of sample (a total of 2 µg) in 1% (v/v) formic acid were loaded with a constant flow of 4 µl/min onto an Acclaim PepMap100 C18 column (0.3 mm id × 5 mm, Dionex Corporation). After trap enrichment, peptides were eluted onto an Easy-Spray PepMap C18 nano column (75 µm × 50 cm, Dionex Corporation) with a linear gradient of 5–35% solvent B (90% acetonitrile with 0.1% formic acid) over 240 min with a constant flow of 200 nl/min. The HPLC system was coupled to an OrbiTrap QExactive mass spectrometer (Thermo Fisher Scientific Inc) via an EasySpray source. The spray voltage was set to 2.0 kV and the temperature of the column set to 40 °C. Full scan MS survey spectra (m/z 350–1600) in profile mode were acquired in the Orbitrap with a resolution of 70,000 after accumulation of 1,000,000 ions. The ten most intense peptide ions from the preview scan in the Orbitrap were fragmented by collision-induced dissociation (normalized collision energy 35% and resolution of 17,500) after the accumulation of 50,000 ions. Maximal filling times were 250 ms for the full scans and 60 ms for the MS/MS scans. Precursor ion charge state screening was enabled and all unassigned charge states as well as singly, 7 and 8 charged species were rejected. The dynamic exclusion list was restricted to a maximum of 500 entries with a maximum retention period of 40 s and a relative mass window of 10 ppm. The lock mass option was enabled for survey scans to improve mass accuracy. Data were acquired using the Xcalibur software version 4.1. Peptide identification was performed with MaxQuant version 1.6.2.2 software using the *S. pombe* proteome from Uniprot.

### Reporting summary

Further information on research design is available in the Nature Portfolio Reporting Summary linked to this article.

## Data availability

The data from the Rpa2-HTP CRAC analysis have been deposited in NCBI's Gene Expression Omnibus and are accessible through GEO Series accession number GSE212930. The mass spectrometry proteomics data was deposited to the ProteomeXchange Consortium via the PRIDE partner repository with the data set identifier PXD037768.

## Code availability

All scripts used for data processing and statistical analysis of CRAC data were written in Python, Perl, or R, and are available on GitHub (https://github.com/TurowskiLab/trxtools).

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

## Acknowledgements

This work was supported by funding from the Natural Sciences and Engineering Research Council of Canada (NSERC) to F.B. (RGPIN-2017-05482). F.B. holds a Canada Research Chair in Quality Control of Gene Expression. C.Y.-S. was supported by the Fonds de recherche du Québec—Santé (FRQS). Work by T.W.T. is supported by the Polish National Agency for Academic Exchange (PPN/PPO/2020/2/00004/U/00001).

## Author contributions

M.D. and F.B. conceived the study and the experimental frame. M.D. prepared the strains, performed TurboID, ChIP assays, and RNA analyses with help from C.Y.-S.. E.P. performed the CRAC assay and T.W.T. analyzed the CRAC data with input from F.B. and D.T.. F.B. wrote the manuscript together with M.D., which was reviewed by all authors.

## Competing interests

The authors declare no competing interests.
