## [Peer Review File · Nature Communications]

The conserved RNA-binding protein Seb1 promotes cotranscriptional ribosomal RNA processing by controlling RNA polymerase I progressionREVIEWER COMMENTS

Reviewer #1 (Remarks to the Author):

In this study, the authors identify a prominent interaction between the yeast protein Seb1 and the ribosomal DNA. Using a series of solid genetic depletion and complementation assays, they reveal that loss of Seb1 or mutation of the RRM motif in Seb1 induces rRNA processing defects. Analysis of RNA polymerase occupancy (ChIP and CRAC) suggested a model that perhaps loss of Seb1 induces faster transcription elongation by RNA polymerase I, and defective coupling of rRNA processing.

Overall, the model is interesting, as it could reveal an additional similarity between RNA polymerases I and II. The rRNA processing defects are clear and compelling, as are the data suggesting Seb1 association with rDNA. However, the implication of RNA polymerase I transcription elongation rate is not compelling to this reviewer. It is challenging to draw conclusions about reaction rates when there are limited or no kinetic analyses. The study would benefit from kinetic analyses *in vivo* like those published previously by the Tollervey lab.

1. If Rrn3 occupancy does not change, but RNAPI occupancy is decreased in Seb1-depletion, one cannot uniquely conclude that transcription initiation is not affected. This occupancy comparison is the main piece of data used to conclude that transcription elongation is enhanced in the mutant cells.
2. Figure 3: How is calculating standard deviation of $n=2$ justified? This is just the range between the data.
3. The differences observed in the CRAC data (figure 5) are not very large. The range between replicates is shown, but the effects appear quite modest. More extensive quantification/analyses are required.
4. To support a pausing role for Seb1, could the authors overexpress the protein and evaluate RNA polymerase I occupancy?

Reviewer #2 (Remarks to the Author):

The manuscript entitled "The conserved RNA-binding protein Seb1 promotes cotranscriptional ribosomal RNA processing by controlling RNA polymerase I progression" by Maxime Duval et al establish a novel function of Seb1 in controlling rRNA production *in vivo*. Authors could show a clear binding to rDNA and rRNA of Seb1 using respectively Chip-seq and CRAC. Affinity purification and proximity-dependent biotinylation coupled to MS were used to explore interactome of Seb1, and confirms a clear interaction with RNAPI machinery and ribosome assembly factors. Upon Seb1 depletion, RNAPI recruitment to rDNA is affected, but not its initiation factor *rrn3*, suggesting a functional link between Seb1 and rRNA production during elongation. Careful quantification of rRNA production by pulse-chase experiment following Seb1 deletion show an early processing defect, indicative of delay cleavage at site A2. Importantly, Seb1 allele defective in CTD binding to RNAPII are not affected in rRNA processing, proving a novel function of Seb1, independent of its binding to RNAPII CTD. Finally, authors could show using RNAPI CRAC that Seb1 promote pausing during transcription.

Altogether, this work clearly demonstrate a novel function of Seb1 in regulating RNAPI elongation, and promote efficient rRNA processing. Similar conclusions have already been proposed for RNAPII factors such as Spt4/spt5, TFIIH, Ctk1 or Paf1. However, for most of those factors, it is virtually impossible to show that RNAPI effect is independent of known RNAPII. In this study, Seb1 allele affecting RNAPII binding have no effect on RNAPI pausing and rRNA processing. Therefore, Seb1 function in RNAPI pausing is novel and conclusion are fully supported by data presented. Seb1 study might open the way

for more mechanistic study on RNAPI elongation regulation.

Major comments

RNAPI activity is exquisitely regulated according to growth rate. It is very important to establish that RNAPI pausing observed here is not due to indirect effect caused by growth rate modification during Seb1 depletion. Authors have depleted Seb1 using two independent system, auxin degron and thiamin repression. Following 12 hours of thiamine depletion, RNAPI recruitment is affected (see Figure 2B). Authors should document in supplementary data growth rate under those depletion condition. Ideally, a depletion of an unrelated factors could be provided as control to establish that RNAPI recruitment defect is not an indirect effect of growth delay. Along the same line, is anchor away procedure for 2 hours of seb1 affecting growth rate ? Unrelated anchor away could have been used here as control for ChIP of RNAPI.

Authors should show if RNAPI interaction is direct, via protein-protein interaction, or mediated by nascent rRNA. IP, and ChIP analysed using RT-qPCR should be performed with or without RNase treatment prior to purification to evaluate the possible contribution of RNA bridging the interaction to rDNA.

Pulse chase experiment and Northern blot indicate a delay cleavage at site A2. In pulse experiment, 20S production appears delay, but is not directly analyzed by Northernblot. Authors should evaluate 20S abundance, and short rRNA product generated from 5'ETS to evaluate A0 and A1 cleavage efficiency (fragment TSS-A0, TSS-A1, A0-A1).

Seb1-RRM MUT allele seems to have a dominant negative effect on rRNA processing, as shown by the higher 35S/27S ratio than seb1 depletion alone. Seb1-RRM mutant could have been used to show that pausing was even prevented.

Seb1 binding to rDNA or rRNA using Chip seq and Crac analysis of respectively, reveal a surprisingly different interaction, with an apparent depletion in ChIP signal were CRAC reads are the highest. Authors have performed ChIP analysed by qPCR using RPA2-HTP. Authors should present how this surprising results fit with model presented in Figure6, by indicating how Seb1 might be first recruited to pre-ribosome without a direct contact to RNA (ChIP), and next binding to rRNA at specific site (CRAC).

Minor comments :

Supplementary table S1: all MS analysis, and all PDB-MS could have been provided in the table, and not only the 268 common to both. Some quality criteria (peptide coverage? Counts??) could also be include to document the proteomic analysis performed here. For all co-purified protein, systematic ID should be provided. Gene name can be misleading (nuc1 for the largest RNAPI subunits, called with synonyms - rpa1 in the excel), SPBC28F2.11 is hmo1 in Pombase. SPCC4B3.08 is Lsg1 complex gamma subunit Lsg1. Interestingly, orthologs in cerevisiae, Ctk1 was involved in regulating RNAPI activity (Bouchoux et al., NAR 2004 PMID: 15520468 ; Grenetier, NAR, 2006 PMID: 16984969)

Some modification of Figure 1 would improve readability : position in basepair of key position would be useful. Panel C depict the different element in one unit of rDNA, but Panel A and H depict position of 18S, 5.8S and 25S, but it is unclear if 5'ETS and 3'ETS are also present in the graph. One scale bar depicting 1kb (panel C) or 5 kb (panel A) is not sufficient to identify position of different element (namely 5'ETS, 3'ETS and IGS). Authors report a binding of Seb1 by CRAC to ITS1 and 2, and 3'ETS. Does Seb1 bind to 5'ETS?

rDNA copies number variation can affect ChIP efficiency. rDNA copies number variation in wild-type *S. cerevisiae*, or in various human cell line are frequently observed. Since rDNA copies number variation

was also documented in some *S. pombe* mutant, authors should rule out that rDNA copies number is modified relative to isogenic WT in strain bearing Pnmt1-seb1, or seb1-AA constructs (using either PFGE, qPCR or ddPCR).

In supplementary figure S2C: is the decrease ChIP signal significant? Authors should perform statistical test between replicate to establish, or not this apparent difference.

Figure 4 B vs C: Visual inspection of Northern between line 1 and 3 indicate large variation in 35S relative to 27S, which are not visible in quantification (line 1 vs 3 in panel C). Furthermore, The error bar is not visible for line 3. Please clarify this point.

Page 10, "Region with high signal are interpreted as having high RNAPII..." – "" valleys reflect low RNAPII occupancy...". Authors should clarify if they want to mentioned published RNAPII CRAC, or should correct this statement to RNAPI.

Higher eukaryotes refer to both plant and metazoan, and is not very precise evolutive. I would rather favor metazoan to describe conservation between yeast and animals

Reviewer #3 (Remarks to the Author):

The manuscript by Duval et al., hones in on an additional role of seb1 in *S.pombe*, namely as a RNA polymerase I interactor regulating rRNA processing. The following study should be considered in the context of publications by the same lab (2016, ref. 19) and the Vasiljeva (2017, ref. 20) and Madhani groups (2018, ref. 21) describing seb1 as a facilitator of RNA 3'end processing and mediator of RNAPII transcription termination and long-lived transcriptional pausing events.

Here, the authors find that seb1 associates with half of the RNAPI subunits and many related factors, as identified by two independent proteomics' strategies. Furthermore, the authors reveal that seb1 is deposited throughout rDNA loci (identified by ChIP-seq) and binds to rRNA (CRAC). The authors show very convincingly that the processing of the rRNA is impaired in the absence of seb1. They go on to correlate impaired rRNA processing with increased transcriptional elongation of the rDNA, as read out by the reduced coverage of rpa2 CRAC signal throughout the rRNA (except for the 5' and 3' end). The strengths of the study lie in the authors' in-depth knowledge of seb1 based on previous work, the availability of seb1's ChIP-seq data, seb1 mutants that can elegantly separate the RNA-binding ability from the RNAPII complex association and the high quality rpa2 CRAC data.

While the execution of the experiments and quality of the data is very high, aspects related to the novelty could be broadened and the interpretation of some of the data could be further strengthened to engage the broad Nature Communication readership. Major and minor concerns are outlined below:

Major points

Figure 1: The authors hypothesize that seb1 associates with rDNA directly (p.12; discussion section), thereby interferes with the elongation speed of RNAPI, causing pausing that is required for the co-transcriptional RNA processing. One caveat is that the recovery of formaldehyde-based crosslinking sites of seb1 ChIP-seq data could also originate from the recovery of co-crosslinked RNAPI subunits, considering the interactors that have been recovered in Fig.1D-F. Therefore, the direct nature of the binding would need to be validated with an alternative strategy or the claim should be removed from the paper. One alternative hypothesis could be that seb1 causes the assembly of a different RNAPI complex leading to changes in transcriptional elongation speed.

Figure 4: The authors show that seb1 can bind or at least crosslink to DNA, RNA and the RNAPI complex (Figure 1). However, when it comes to explaining seb1's mode of action as rRNA-processing regulator, the authors only disrupt the binding of seb1 to RNA to prove its relevance. To streamline the narrative and clarify which of these roles is necessary or required to act as a rRNA processing

regulator, the authors need to generate additional *seb1* mutants to differentiate between the contribution of binding to the RNAPI complex, binding to DNA and RNA (similarly to the publication from 2016).

Figure 3 and 4: What is the experimental difference behind the experiments in Fig. 3A and 4B? Why are the effects a lot bigger for Figure 3A especially when comparing the Northern blot images? Does the efficiency of *seb1* depletion vary considerably between these experiments? Do the authors have some way of assessing the efficiency of *seb1* depletion upon addition of thiamine here? The authors refer back to the validation experiments of the *Pnmt1-seb1* strain that have previously been performed by the same group (ref. 19). It would be advantageous to show that the depletion is functional given the conditions in this manuscript to ensure that the results are based on the loss of *seb1*.

Figure 5: The peaks of the *rpa2* CRAC experiment in the *seb1*⁻-deficient strain appear to be consistently reduced, except at—the authors refer to as—the initiation-stalled site and upstream of the termination-stalled site. The assumption that the reduction in the *rpa2* coverage is a consequence of different speeds in RNAPI's transcriptional elongation is not a very clean line of argumentation. In addition, the authors detect deposition of *seb1* on the rRNA at very specific sites (Fig. 1G). However, the *seb1* ChIP data does not show this "pausing" pattern. It would be helpful, if the authors addressed this discrepancy (technical limitation of ChIP?). In order to support the claim that *seb1* causes differences in RNAPI's transcriptional elongation speed, the authors should verify this with a more direct approach, e.g. SLAMseq (PMID: 35296539; PMID: 28945705).

Figure 6: The model builds on existing proof that a change of transcriptional speed causes changes in co-transcriptional processing (as seen in splicing and 3'end processing). However, the authors do not formerly prove that *seb1* impacts rRNA processing co-transcriptionally (as claimed in the title). Most of the data related to that aspect are circumstantial or inferences from the available literature. In order to at least in part prove that the *seb1*-mediated rRNA processing takes place co-transcriptionally, the generation of a loss-of-RNAPI-association mutant in combination with the Northern blotting analysis would be required.

Novelty claim: While the authors nicely expand the already known functions of *seb1*, the models for the RNAPII- (as claimed by ref. 19) and RNAPI-associated roles are very similar without providing a more in-depth analysis of what that might mean for the cellular state. Therefore, expanding on any of the following questions would provide an additional novel viewpoint, which remained unexplored thus far: What drives *seb1*'s association with RNAPI and/or RNAPII? Is it a stochastic effect or is there a specific regulation? Is *seb1* equally likely to associate with either of the polymerases? Are there any physiological conditions where the association with one complex is favoured over the other or where *seb1* is dramatically differentially expressed?

Minor Points:

Fig. 1B: Is it expected that the binding of *seb1* varies across the rDNA? What is the reason for the accumulation at position 5? It would be helpful, if the authors discussed these data in a bit more detail in the text.

Fig. 1H: Would it be possible to depict the two biological replicates separately to assess the reproducibility of the binding of *seb1* to the selected rRNA locus?

Fig. 2B and C: Please, indicate which of the differences are statistically significant.

Reviewer #4 (Remarks to the Author):

According to my expertise in clinical proteomics I've been contacted to specifically review the

proteomics parts of the manuscript entitled "The conserved RNA-binding protein Seb1 promotes cotranscriptional ribosomal RNA processing by controlling RNA polymerase I progression" submitted by Duval et al. After carefully revising this aspect alongside the manuscript I would like to say that the AP-MS and PDB-MS strategies used in this study are challenging strategies that involve an intensive sample processing and subsequent liquid chromatography coupled to mass spectrometry analysis and authors performed them magisterially obtaining short lists of interacting proteins. Nonetheless, when revising the Methods section I noticed that an specific section for the proteomics identification of proteins by mass spectrometry is totally missing in methods section. The details of the sample processing performed for the biotin-labeled enrichment is included in the TurboID section, but other details about the sample processing and instrument configuration are missing. In my opinion both studies, PDB-MS and AP-MS require further details in Methods section in an independent section. Currently, the only information provided about the detection of peptides is that sentence "Peptide identification was performed with MaxQuant software using the *S. pombe* proteome from Uniprot.". Details about sample preparation for AP and PDB, chromatographic separation of peptides, mobile phases, gradient, column, and instrument parameters and configuration, such as error tolerance of parent and fragments, FDR, collision mode, etc. are mandatory when including proteomics experiments in a manuscript.

We have addressed all the very useful and constructive comments from the four reviewers. Specific responses to the individual concerns are included below and addressed, point-by-point. We have also included a marked-up version of our revised manuscript in which all the specific changes introduced to the text are highlighted in red.

Reviewer #1

In this study, the authors identify a prominent interaction between the yeast protein Seb1 and the ribosomal DNA. Using a series of solid genetic depletion and complementation assays, they reveal that loss of Seb1 or mutation of the RRM motif in Seb1 induces rRNA processing defects. Analysis of RNA polymerase occupancy (ChIP and CRAC) suggested a model that perhaps loss of Seb1 induces faster transcription elongation by RNA polymerase I, and defective coupling of rRNA processing.

Overall, the model is interesting, as it could reveal an additional similarity between RNA polymerases I and II. The rRNA processing defects are clear and compelling, as are the data suggesting Seb1 association with rDNA. However, the implication of RNA polymerase I transcription elongation rate is not compelling to this reviewer. It is challenging to draw conclusions about reaction rates when there are limited or no kinetic analyses. The study would benefit from kinetic analyses in vivo like those published previously by the Tollervey lab.

RESPONSE: We thank the Reviewer for his/her interest in the study and the valuable comments. The specific concerns/questions of the Reviewer are addressed below.

The study would benefit from kinetic analyses in vivo like those published previously by the Tollervey lab.

RESPONSE: The exhaustive kinetic analyses established by the Tollervey lab in 2010 (Kos et al., Mol Cell), which involved harvesting of cells at 10 seconds intervals, was used to demonstrate that 70% of nascent pre-rRNAs undergo co-transcriptional cleavage in yeast. Our current study reveals that Seb1 is required for optimal co-transcriptional cleavage of pre-rRNAs. This conclusion is supported by (i) Northern blot analyses indicating that the co-transcriptional cleavage efficiency at site A₂ was significantly decreased in Seb1-deficient cells, leading to an increase in the 35S/27A₂ ratio and (ii) rRNA pulse-chase assays showing delayed kinetics in processing the 35S pre-rRNA into 27S and 20S precursors (Figure 3). Although we agree with the review that extensive high resolution kinetic labeling in vivo would provide further quantitative details not accessible in our rRNA pulse-chase assays, such data would likewise confirm the main conclusion that the absence of Seb1 impairs co-transcriptional pre-rRNA processing, which is already supported by our Northern blot and rRNA pulse-chase assays. We therefore acknowledge the reviewer's suggestion for sampling metabolically labeled yeast cells at 10-15s intervals to obtain more quantitative insights into the consequence of Seb1 deficiency on co-transcriptional pre-rRNA processing, but believe that the addition of such kinetic assays are not critically required to support the main conclusions reported in the current study and that detailed kinetic assays could be included in future work.

1. If Rrn3 occupancy does not change, but RNAPI occupancy is decreased in Seb1-depletion, one cannot uniquely conclude that transcription initiation is not affected. This occupancy comparison is the main piece of data used to conclude that transcription elongation is enhanced in the mutant cells.

RESPONSE: Our manuscript includes several pieces of evidence that support that RNAPI progression is enhanced in Seb1-deficient cells: i) Rrn3 occupancy does not change, (ii) reduced RNAPI occupancy, and (iii) decreased RNAPI pausing shown by CRAC. We agree with the reviewer that one cannot uniquely conclude that RNAPI progression is enhanced based on the ChIP assays showing the lack of change in Rrn3 occupancy and the reduced RNAPI density. Accordingly, this is why we performed the CRAC assays shown in Figure 5 to assess the distribution of RNAPI at the single nucleotide resolution in wild-type and Seb1-deficient cells, which revealed that Seb1 deficiency leads to decreased pausing/slow states during RNAPI transcription elongation.

2. Figure 3: How is calculating standard deviation of n=2 justified? This is just the range between the data.

RESPONSE: Good point. With two independent biological replicates, it does not make sense to graph a mean with a standard deviation. Accordingly, Figure 3d now shows the graph with the two independent values (and the mean) for each time point and the computed t-tests. Only two independent experiments were done for the rRNA pulse-chase assays as these are relatively complicated experiments and the differences between wild-type and *Seb1*-deficient cells were clear. We thank the reviewer for noting this point.

3. The differences observed in the CRAC data (figure 5) are not very large. The range between replicates is shown, but the effects appear quite modest. More extensive quantification/analyses are required.

RESPONSE: As recommended by the reviewer, we have now extended our CRAC analysis further. First, we calculated log₂ ratios of the sequencing coverage for the ITS1-5.8S-ITS2 region, which showed 2- to 4-fold variations in RNAPI presence between the wild-type control and the *seb1* mutant. This new data is presented in Figure 5c and described on p. 11 (lines 22-25) of the revised manuscript. Furthermore, average RNAPI transcription speed was used to transform fraction of reads to occupancy time in seconds. Notably, this analysis revealed that in the absence of *Seb1*, the average occupancy time of RNAPI dropped substantially, from 150ms in wild-type cells to 50ms in *Seb1*-deficient cells at specific positions in the ITS1-5.8S-ITS2 region. This new analysis is described on p. 11 (lines 26-30) and shown in Supplementary Figure S5d of the revised manuscript. We thank the reviewer for the comment, as these new analyses strengthen the conclusion that the rate of RNAPI progression varies considerably at specific rDNA regions between wild-type and *Seb1*-deficient cells.

4. To support a pausing role for *Seb1*, could the authors overexpress the protein and evaluate RNA polymerase I occupancy?

RESPONSE: This is an excellent idea, but it turns out that overexpressing *Seb1* is quite challenging. As suggested by the reviewer, we inserted a transgenic copy of the *seb1* gene under the control of the thiamine-sensitive *nmt1* promoter. We found that removal of thiamine to induce the transgenic copy of *seb1* rapidly resulted in growth arrest, suggesting that excess *Seb1* is toxic to fission yeast. Consistent with this idea, we also found that the excess transgenic *Seb1* negatively controlled the expression of the endogenous copy of the *seb1* gene. As *Seb1* is one of the most abundant proteins in fission yeast (almost 20,000 copies/cell: PMID: 24763107), *S. pombe* does not appear to tolerate excess *Seb1*, which restrained us to assess whether excess *Seb1* affected RNAPI pausing.

Reviewer #2

The manuscript entitled “The conserved RNA-binding protein Seb1 promotes cotranscriptional ribosomal RNA processing by controlling RNA polymerase I progression” by Maxime Duval et al establish a novel function of Seb1 in controlling rRNA production in vivo. Authors could show a clear binding to rDNA and rRNA of Seb1 using respectively Chip-seq and CRAC. Affinity purification and proximity-dependent biotinylation coupled to MS were used to explore interactome of Seb1, and confirms a clear interaction with RNAPI machinery and ribosome assembly factors. Upon Seb1 depletion, RNAPI recruitment to rDNA is affected, but not it's initiation factor *rrn3*, suggesting a functional link between Seb1 and rRNA production during elongation. Careful quantification of rRNA production by pulse-chase experiment following Seb1 deletion show an early processing defect, indicative of delay cleavage at site A2. Importantly, Seb1 allele defective in CTD binding to RNAPII are not affected in rRNA processing, proving a novel function of Seb1, independent of it's binding to RNAPII CTD. Finally, authors could show using RNAPI CRAC that Seb1 promote pausing during transcription.

Altogether, this work clearly demonstrate a novel function of Seb1 in regulating RNAPI elongation, and promote efficient rRNA processing. Similar conclusions have already been proposed for RNAPII factors such as Spt4/spt5, TFIIH, Ctk1 or Paf1. However, for most of those factors, it is virtually impossible to show that RNAPI effect is independent of known RNAPII. In this study, Seb1 allele affecting RNAPII binding have no effect on RNAPI pausing and rRNA processing. Therefore, Seb1 function in RNAPI pausing is novel and conclusion are fully supported by data presented. Seb1 study might open the way for more mechanistic study on RNAPI elongation regulation.

RESPONSE: We would like to thank this reviewer for the constructive comments and enthusiasm for the study. This reviewer raised several excellent questions that are addressed below.

MAJOR COMMENTS

-RNAPI activity is exquisitely regulated according to growth rate. It is very important to establish that RNAPI pausing observed here is not due to indirect effect caused by growth rate modification during Seb1 depletion. Authors have depleted Seb1 using two independent system, auxin degron and thiamin repression. Following 12 hours of thiamine depletion, RNAPI recruitment is affected (see Figure 2B). Authors should document in supplementary data growth rate under those depletion condition. Ideally, a depletion of an unrelated factors could be provided as control to establish that RNAPI recruitment defect is not an indirect effect of growth delay. Along the same line, is anchor away procedure for 2 hours of *seb1* affecting growth rate? Unrelated anchor away could have been used here as control for ChIP of RNAPI.

RESPONSE: The reviewer raises an excellent point. We have in fact already examined RNAPI occupancy in a different anchor away strain (*Pac1-AA*) in which rapamycin supplementation causes growth arrest (see panel A on the left) and found no significant decrease in RNAPI occupancy after relocalization of *Pac1* to the cytoplasm (see panel B on the left). These data were published in 2020 in NAR (PMID: 34352089) and argue against the idea that reduced growth rate automatically results in reduced RNAPI occupancy. In addition, and as suggested by this reviewer, we have also analyzed RNAPI density by ChIP assays after thiamine-dependent depletion of an unrelated essential protein (*Cbp80*). In contrast to the depletion of *Seb1*, the depletion of *Cbp80* after 12h of thiamine supplementation did not result in reduced RNAPI occupancy, therefore strengthening the conclusion that the reduced RNAPI occupancy shown after *Seb1* depletion is not the indirect consequence of

reduced growth rate. These additional data are now presented as Supplementary Fig. S2a and described on p. 8 (lines 4-7) of the revised manuscript.

-Authors should show if RNAPI interaction is direct, via protein-protein interaction, or mediated by nascent rRNA. IP, and ChIP analysed using RT-qPCR should be performed with or without RNase treatment prior to purification to evaluate the possible contribution of RNA bridging the interaction to rDNA.

RESPONSE: To address whether RNA-protein interactions are required for the recruitment of Seb1 to rDNA during RNAPI transcription, we took advantage of a version of Seb1 with key substitutions in its RNA recognition motif (RRM) that disrupts the ability of Seb1 to bind RNA (PMID: 27401558). Analysis of Seb1 occupancy at rDNA by ChIP assays revealed a striking reduction in recruitment for the Seb1 RRM mutant compared to the wild-type version of Seb1. Although this does not exclude the possibility of protein-protein contacts between Seb1 and specific subunits of the RNAPI complex, these results indicate that interactions between Seb1 and nascent pre-rRNA are required for the stable recruitment of Seb1 to rDNA. These new results are now presented in Supplementary Fig. S1b and described on p. 6 (lines 8-12) of the revised manuscript.

-Pulse chase experiment and Northern blot indicate a delay cleavage at site A2. In pulse experiment, 20S production appears delay, but is not directly analyzed by Northernblot. Authors should evaluate 20S abundance, and short rRNA product generated from 5'ETS to evaluate A0 and A1 cleavage efficiency (fragment TSS-A0, TSS-A1, A0-A1).

RESPONSE: We had performed several attempts to include data on 20S pre-rRNA abundance in the original manuscript, but were unfortunately not successful at detecting this precursor products by Northern. We tried again during the revision, but without much success. The ITS1-specific sequences downstream of the 18S rRNA are highly AT-rich in *S. pombe* and we think that this impedes with our ability to specifically detect 20S rRNA precursors using a labeled short DNA probe. We were, however, able to examine the steady state levels of short RNA products generated from the 5'ETS to assess the impact of Seb1 deficiency on A₀ and A₁ cleavage efficiency. Northern analysis of short RNAs produced from 5'ETS processing revealed reduced cleavage at A₀ and A₁ in the *seb1* mutant, which is consistent with the accumulation of 35S, 33S, and 32S pre-rRNAs detected in Seb1-deficient cells (Fig. 3a). These additional results are presented as Supplementary Fig. S3d-S3e and described on p. 9 (lines 7-9) of the revised manuscript. Given that sites A₀ and A₁ within the 5'ETS, and A₂ in ITS1, are primarily cleaved co-transcriptionally in yeast, the reduced cleavage efficiency at sites A₀, A₁, and A₂ in Seb1-deficient cells are consistent with our model whereby Seb1-dependent changes in RNAPI elongation rates impair the efficiency of co-transcriptional rRNA processing (Fig. 6). We thank the reviewer for suggesting this additional analysis as it strengthens our study.

-Seb1-RRM MUT allele seems to have a dominant negative effect on rRNA processing, as shown by the higher 35S/27S ratio than *seb1* depletion alone. Seb1-RRM mutant could have been used to show that pausing was even prevented.

RESPONSE: Although the mean value of 35S/27S ratio is slightly greater for the Seb1-RRM mutant compared to the empty vector control, the data are not significantly different between these two strains based on statistical analysis. Accordingly, we do not believe that the RRM mutant allele of *seb1* acts in a dominant-negative manner. We have now indicated that these data are not significantly different in Figure 3c of the revised manuscript.

-Seb1 binding to rDNA or rRNA using Chip seq and Crac analysis of respectively, reveal a surprisingly different interaction, with an apparent depletion in ChIP signal were CRAC reads are the highest. Authors have performed ChIP analysed by qPCR using RPA2-HTP. Authors should present how this surprising results fit with model presented in Figure6, by indicating how Seb1 might be first recruited to pre-ribosome without a direct contact to RNA (ChIP), and next binding to rRNA at specific site (CRAC).

RESPONSE: The reviewer likely refers to the lower levels of Seb1 cross-linking in ITS1 and ITS2 regions of the rDNA compared to the coding regions of the rDNA as detected by ChIP-seq in Fig. 1a. The reduced levels of ChIP-seq signal in ITS1 and ITS2 regions are in fact generalized to most of our ChIP-seq analysis, as this reduction was also observed for Dhp1 and Cbf5 proteins. In addition, analysis of DNA content of starting chromatin extracts (input) revealed reduced signal in ITS1 and ITS2 regions (see below), suggesting that these chromatin regions are less accessible (soluble?) for

copurification by ChIP assays after formaldehyde crosslinking, thereby explaining the generalized depletion of ChIP signal in ITS1 and ITS2 in Figure 1a. We have now added this sentence on p. 6 (lines 16-19) of the revised manuscript: “*The generalized depletion of ChIP-seq signal in ITS1 and ITS2 regions of the rDNA gene appears to be the consequence of reduced DNA content for those regions in the starting chromatin extracts*” to explain for the generalized depletion of ChIP signal in ITS1 and ITS2 in Figure 1a.

MINOR COMMENTS :

-Supplementary table S1: all MS analysis, and all PDB-MS could have been provided in the table, and not only the 268 common to both. Some quality criteria (peptide coverage? Counts??) could also be included to document the proteomic analysis performed here. For all co-purified protein, systematic ID should be provided. Gene name can be misleading (nuc1 for the largest RNAPI subunits, called with synonyms - rpa1 in the excel), SPBC28F2.11 is hmo1 in Pombase. SPCC4B3.08 is Lsg1 complex gamma subunit Lsg1. Interestingly, orthologs in cerevisiae, Ctk1 was involved in regulating RNAPI activity (Bouchoux et al., NAR 2004 PMID: 15520468 ; Grenetier, NAR, 2006 PMID: 16984969)

RESPONSE: We agree with the reviewer. For the affinity purification coupled to MS (AP-MS) analysis of Seb1, all the proteomic data (peptide and MS counts, sequence coverage, peptide intensities, etc) are provided in detail in a previously published paper by Lemay et al., Genes Dev 2016 PMID 27401558. For the new PDB-MS analysis of Seb1 that accompanies the current study, we have revised the manuscript by adding Supplementary Table S1, which includes MS data such as MS counts, unique peptides, amino acid sequence coverage, and LFQ intensities for control and Seb1-TurboID strains. This supplementary Table is cited on p. 6 (lines 29-30) of the revised manuscript. As suggested by the reviewer, Supplementary Table S1 also includes systematic IDs in the Fasta headers column.

Some modification of Figure 1 would improve readability : position in basepair of key position would be useful. Panel C depict the different element in one unit of rDNA, but Panel A and H depict position of 18S, 5.8S and 25S, but it is unclear if 5'ETS and 3'ETS are also present in the graph. One scale bar depicting 1kb (panel C) or 5 kb (panel A) is not sufficient to identify position of different element (namely 5'ETS, 3'ETS and IGS). Authors report a binding of Seb1 by CRAC to ITS1 and 2, and 3'ETS. Does Seb1 bind to 5'ETS?

RESPONSE: As suggested by the reviewer, we have now revised Figure 1 and included 5'ETS and 3'ETS regions to panels (a) and (h). The revised version of the Seb1 CRAC analysis presented in Fig. 1h shows binding of Seb1 to 5'ETS sequences; yet, this signal is less compared to ITS and 3'ETS regions of the pre-rRNA.

rDNA copies number variation can affect ChIP efficiency. rDNA copies number variation in wild-type *S. cerevisiae*, or in various human cell line are frequently observed. Since rDNA copies number variation was also documented in some *S. pombe* mutant, authors should rule out that rDNA copies number is modified relative to isogenic WT in strain bearing *Pnmt1-seb1*, or *seb1-AA* constructs (using either PFGE, qPCR or ddPCR).

RESPONSE: As suggested by the reviewer, we have compared rDNA copy number between wild-type and *Pnmt1-seb1* strains using qPCR. This analysis showed no significant changes in rDNA copy number in Seb1-deficient cells compared to an isogenic wild-type strain. As a control, we observed a

significant reduction in rDNA copy number in cells deleted for *dcr1*, as previously described (PMID: 25417108). These data are now presented as Supplementary Fig. S2b and described on p. 8 (lines 7-8) of the revised manuscript.

Figure 4 B vs C: Visual inspection of Northern between line 1 and 3 indicate large variation in 35S relative to 27S, which are not visible in quantification (line 1 vs 3 in panel C). Furthermore, The error bar is not visible for line 3. Please clarify this point.

RESPONSE: The analysis of 35S/27S ratio by Northern blot assays shown in Figure 4 are from three independent biological replicates. The quantification of Northern blot data shown in Fig. 4c are expressed relative to the wild-type *Seb1*-Flag construct (lane 3) and this is why error bars are not shown for this strain. Statistical analysis of the data did not show significant difference between the untagged control strain (lane 1) and *Seb1*-deficient cells that expressed the wild-type *Seb1*-Flag construct (lane 3).

-Page 10, “Region with high signal are interpreted as having high RNAPII...” – “” valleys reflect low RNAPII occupancy...”. Authors should clarify if they want to mentioned published RNAPII CRAC, or should correct this statement to RNAPI.

RESPONSE: We thank the reviewer for noting this oversight. We have corrected this statement to mention RNAPI in the revised manuscript.

-Higher eukaryotes refer to both plant and metazoan, and is not very precise evolutive. I would rather favor metazoan to describe conservation between yeast and animals

RESPONSE: We agree. We have made the correction on p. 16 (line 16).

Reviewer #3

The manuscript by Duval et al., hones in on an additional role of seb1 in *S.pombe*, namely as a RNA polymerase I interactor regulating rRNA processing. The following study should be considered in the context of publications by the same lab (2016, ref. 19) and the Vasiljeva (2017, ref. 20) and Madhani groups (2018, ref. 21) describing seb1 as a facilitator of RNA 3' end processing and mediator of RNAPII transcription termination and long-lived transcriptional pausing events. Here, the authors find that seb1 associates with half of the RNAPI subunits and many related factors, as identified by two independent proteomics' strategies. Furthermore, the authors reveal that seb1 is deposited throughout rDNA loci (identified by ChIP-seq) and binds to rRNA (CRAC). The authors show very convincingly that the processing of the rRNA is impaired in the absence of seb1. They go on to correlate impaired rRNA processing with increased transcriptional elongation of the rDNA, as read out by the reduced coverage of rpa2 CRAC signal throughout the rRNA (except for the 5' and 3' end). The strengths of the study lie in the authors' in-depth knowledge of seb1 based on previous work, the availability of seb1's ChIP-seq data, seb1 mutants that can elegantly separate the RNA-binding ability from the RNAPII complex association and the high quality rpa2 CRAC data. While the execution of the experiments and quality of the data is very high, aspects related to the novelty could be broadened and the interpretation of some of the data could be further strengthened to engage the broad Nature Communication readership. Major and minor concerns are outlined below:

We thank Reviewer #3 for his/her interest in the study and the valuable comments. The specific concerns/questions of the Reviewer are addressed below.

MAJOR POINTS

-Figure 1: The authors hypothesize that seb1 associates with rDNA directly (p.12; discussion section), thereby interfering with the elongation speed of RNAPI, causing pausing that is required for the co-transcriptional RNA processing. One caveat is that the recovery of formaldehyde-based crosslinking sites of seb1 ChIP-seq data could also originate from the recovery of co-crosslinked RNAPI subunits, considering the interactors that have been recovered in Fig.1D–F. Therefore, the direct nature of the binding would need to be validated with an alternative strategy or the claim should be removed from the paper. One alternative hypothesis could be that seb1 causes the assembly of a different RNAPI complex leading to changes in transcriptional elongation speed.

RESPONSE: Our reference to Seb1 acting “directly” on RNAPI transcription in the Discussion section (p. 13) reflected the idea that the observed Seb1-dependent changes in RNAPI dynamics and pre-rRNA processing were unlikely to be caused by indirect (collateral) changes in the expression of RNAPII-expressed protein-coding genes involved in ribosome biogenesis. Seb1 does not have DNA-binding activity and is therefore unlikely to associate with the rDNA directly. As stated on p. 14 (lines 29-33) and p. 15 (lines 1-6), we prefer a model whereby Seb1 is recruited to the rDNA via RNAPI interactions or nascent rRNA, and not via association with the rDNA. We have modified the text on p.13 (lines 13-21) to clarify that we do not hypothesize a direct association of Seb1 with the rDNA, but rather a “direct role” of Seb1 on RNAPI transcription.

Figure 4: The authors show that seb1 can bind or at least crosslink to DNA, RNA and the RNAPI complex (Figure 1). However, when it comes to explaining seb1's mode of action as rRNA-processing regulator, the authors only disrupt the binding of seb1 to RNA to prove its relevance. To streamline the narrative and clarify which of these roles is necessary or required to act as a rRNA processing regulator, the authors need to generate additional seb1 mutants to differentiate between the contribution of binding to the RNAPI complex, binding to DNA and RNA (similarly to the publication from 2016).

RESPONSE: As mentioned for the previous point, we do not favor a model whereby Seb1 is recruited to sites of RNAPI transcription by directly binding to rDNA, but rather via associations with subunits of the RNAPI complex and/or nascent rRNA. Accordingly, a mutant of Seb1 defective in RNA binding shows loss of function in pre-rRNA processing (Fig. 4) and defective recruitment to sites of RNAPI transcription (Fig. S1b). The creation of versions of Seb1 that disrupt binding to the RNAPI complex will require a detailed understanding of the underlying protein-protein contacts between Seb1 and

specific subunits of the RNAPII transcription complex, which could take several years to achieve. We believe that such biochemical insights into the Seb1-RNAPII protein-protein contacts are beyond the scope of the current study and will be the subject of future studies.

Figure 3 and 4: What is the experimental difference behind the experiments in Fig. 3A and 4B? Why are the effects a lot bigger for Figure 3A especially when comparing the Northern blot images? Does the efficiency of *seb1* depletion vary considerably between these experiments? Do the authors have some way of assessing the efficiency of *seb1* depletion upon addition of thiamine here? The authors refer back to the validation experiments of the *Pnmt1-seb1* strain that have previously been performed by the same group (ref. 19). It would be advantageous to show that the depletion is functional given the conditions in this manuscript to ensure that the results are based on the loss of *seb1*.

RESPONSE: We do not agree that the 35S/27S ratios presented in Fig. 3a and Fig. 4b are substantially different. The mean value for the increase in 35S/27S ratio in Fig. 3a-3b is 4.5 (N=6) and 3.1 in Fig. 3b-3c (N=3). These values are not drastically different and can be explained by random variations during the experimental process as well as the number of measured values (N=6 vs N=3). Importantly, we have used a completely independent *Seb1* conditional strain to further show a significant increase in the 35S/27S ratio after depleting *Seb1* from the nucleus (Supplementary Fig. S3b-S3c). Collectively, we believe that the data showing defective pre-rRNA processing in *Seb1*-deficient cells are compelling.

Figure 5: The peaks of the *rpa2* CRAC experiment in the *seb1*-deficient strain appear to be consistently reduced, except at—the authors refer to as—the initiation-stalled site and upstream of the termination-stalled site. The assumption that the reduction in the *rpa2* coverage is a consequence of different speeds in RNAPII's transcriptional elongation is not a very clean line of argumentation. In addition, the authors detect deposition of *seb1* on the rRNA at very specific sites (Fig. 1G). However, the *seb1* ChIP data does not show this “pausing” pattern. It would be helpful, if the authors addressed this discrepancy (technical limitation of ChIP?). In order to support the claim that *seb1* causes differences in RNAPII's transcriptional elongation speed, the authors should verify this with a more direct approach, e.g. SLAMseq (PMID: 35296539; PMID: 28945705).

RESPONSE: First, we would like to stress that we never claimed that the “elongation speed” was altered in *Seb1*-deficient cells, since we did not measure RNAPII speed in the current study. What we show, however, is that RNAPII pausing is less frequent or of shorter duration, thereby leading for a more streamlined progression of RNAPII to reach the 3' end. ChIP does not have the single-nucleotide resolution of CRAC, which examines the 3' end of the nascent RNA in the RNAPII catalytic site. Thus, CRAC allows to assess the frequency of pausing events (peaks), which were found to be reduced in *Seb1*-deficient cells. Second, the initiation stalled-site and termination site are very unique and should not be considered as elongating RNAPII. The initiation site is likely to represent the time needed to clear the promoter and/or dissociation of elongating complex from transcription initiation factors. The termination site is linked with at least two events: stripping of roadblock proteins such as *Nsi1/Reb1* and loss of torsional entrainment. This phenomenon is currently studied by Tollervy and Turowski and a manuscript will be submitted soon. Thus, initiation and termination peaks should not be interpreted as directly related to transcription elongation.

Figure 6: The model builds on existing proof that a change of transcriptional speed causes changes in co-transcriptional processing (as seen in splicing and 3' end processing). However, the authors do not formerly prove that *seb1* impacts rRNA processing co-transcriptionally (as claimed in the title). Most of the data related to that aspect are circumstantial or inferences from the available literature. In order to at least in part prove that the *seb1*-mediated rRNA processing takes place co-transcriptionally, the generation of a loss-of-RNAPII-association mutant in combination with the Northern blotting analysis would be required.

RESPONSE: As mentioned earlier, the generation of *seb1* alleles that disrupt *Seb1* binding to the RNAPII complex will require a detailed understanding of the underlying physical contacts between *Seb1* and specific subunits of the RNAPII transcription complex. To address the reviewer's point about

whether Seb1 contributes to rRNA processing co-transcriptionally, we examined the co-transcriptional recruitment of the Seb1-RRM mutant that is defective in pre-rRNA processing as demonstrated by our Northern blot assays (Fig. 4). Analysis of Seb1 association at the site of rDNA transcription by ChIP assays revealed a striking reduction in recruitment of the Seb1-RRM mutant compared to the wild-type version of Seb1. These new results are now presented in Supplementary Fig. S1b and described on p. 6 (lines 8-12) of the revised manuscript. These results strengthen the idea that Seb1 impacts rRNA processing in a co-transcriptional manner.

Novelty claim: While the authors nicely expand the already known functions of seb1, the models for the RNAPII- (as claimed by ref. 19) and RNAPI-associated roles are very similar without providing a more in-depth analysis of what that might mean for the cellular state. Therefore, expanding on any of the following questions would provide an additional novel viewpoint, which remained unexplored thus far: What drives seb1's association with RNAPI and/or RNAPII? Is it a stochastic effect or is there a specific regulation? Is seb1 equally likely to associate with either of the polymerases? Are there any physiological conditions where the association with one complex is favoured over the other or where seb1 is dramatically differentially expressed?

RESPONSE: To our knowledge, Seb1 is the first RNA-binding protein shown to stimulate pause states for both RNAPI and RNAPII complexes to promote co-transcriptional RNA processing, and as such, this is a novel and significant finding as cellular factors and molecular mechanisms that influence the transcription elongation rate of RNAPI remain poorly understood. Obviously, these new findings prompt additional questions, many of which are mentioned by the reviewer. We are currently investigating some of these important questions in the lab, such as understanding the biochemical nature of the association between Seb1 and the RNAPI and RNAPII transcription machinery as well as whether Seb1 can regulate ribosome biogenesis; yet, this work will be the subject of future studies.

MINOR POINTS:

Fig. 1B: Is it expected that the binding of seb1 varies across the rDNA? What is the reason for the accumulation at position 5? It would be helpful, if the authors discussed these data in a bit more detail in the text.

RESPONSE: The increase in ChIP signal at region 5 in Fig. 1b is not statistically significant compared to regions 4 and 6. Accordingly, the ChIP-seq signal for Seb1 appears relatively homogeneous across the rDNA (Fig. 1a) except for ITS1 and ITS2 regions, which show reduced signal for most of our ChIP-seq analysis, as this depletion is also observed for Dhp1 and Cbf5 proteins. Analysis of DNA content of starting chromatin extracts (input) revealed reduced signal in ITS1 and ITS2 regions (see response to Reviewer #2), suggesting that these chromatin regions are less accessible (soluble?) for copurification by ChIP assays after formaldehyde crosslinking, thereby explaining the generalized depletion of ChIP signal in ITS1 and ITS2 regions in Fig. 1a. As suggested by the reviewer, we have now added sentences on p. 6 (lines 8-12 and 16-19) of the revised manuscript to further discuss the binding profile of Seb1 across the rDNA locus.

Fig. 1H: Would it be possible to depict the two biological replicates separately to assess the reproducibility of the binding of seb1 to the selected rRNA locus?

RESPONSE: The data for the two independent biological replicates of the Seb1 CRAC analysis are now shown in Supplementary Fig. S1c and mentioned on p. 7 (line 19) of the revised manuscript.

Fig. 2B and C: Please, indicate which of the differences are statistically significant.

RESPONSE: As suggested by the reviewer, we have now added statistical analyses for the data shown in Fig. 2b and 2c.

Reviewer #4

According to my expertise in clinical proteomics I've been contacted to specifically review the proteomics parts of the manuscript entitled "The conserved RNA-binding protein Seb1 promotes cotranscriptional ribosomal RNA processing by controlling RNA polymerase I progression" submitted by Duval et al. After carefully revising this aspect alongside the manuscript I would like to say that the AP-MS and PDB-MS strategies used in this study are challenging strategies that involve an intensive sample processing and subsequent liquid chromatography coupled to mass spectrometry analysis and authors performed them magisterially obtaining short lists of interacting proteins. Nonetheless, when revising the Methods section I noticed that a specific section for the proteomics identification of proteins by mass spectrometry is totally missing in methods section. The details of the sample processing performed for the biotin-labeled enrichment is included in the TurboID section, but other details about the sample processing and instrument configuration are missing. In my opinion both studies, PDB-MS and AP-MS require further details in Methods section in an independent section. Currently, the only information provided about the detection of peptides is that sentence "Peptide identification was performed with MaxQuant software using the *S. pombe* proteome from Uniprot.". Details about sample preparation for AP and PDB, chromatographic separation of peptides, mobile phases, gradient, column, and instrument parameters and configuration, such as error tolerance of parent and fragments, FDR, collision mode, etc. are mandatory when including proteomics experiments in a manuscript.

RESPONSE: We agree with the reviewer that details about the peptide sample processing and MS instrument configuration were missing in the original manuscript. As suggested by the reviewer, we have now included an additional paragraph to the Methods section of the manuscript to describe the LC-MS/MS analysis. The detailed MS analysis and instrument parameters can be found on p. 20 (lines 36-48) and p. 21 (lines 1-8) and of the revised manuscript.

REVIEWERS' COMMENTS

Reviewer #1 (Remarks to the Author):

The authors have attempted to address most of the comments raised in the previous round of review, with a notable exception of including kinetic analyses to support their conclusions. However, the revised manuscript seems to put less emphasis of transcription elongation rate versus pausing, which may be appropriate.

As noted by multiple reviewers, it is difficult to deconvolute direct and indirect effects of Seb1 on RNA polymerases I and II. The authors added a discussion of this fact, and they recognize limitations of the genetic and molecular tools employed.

In summary, the work is interesting and will generate meaningful conversation in the field.

Reviewer #2 (Remarks to the Author):

The revised manuscript entitled "The conserved RNA-binding protein Seb1 promotes cotranscriptional ribosomal RNA processing by controlling RNA polymerase I progression" by Maxime Duval et al establish a novel function of Seb1 in controlling rRNA production in vivo. Authors could show a clear binding to rDNA and rRNA of Seb1 using respectively Chip-seq and CRAC.

Altogether, this revised work clearly demonstrate a novel function of Seb1 in regulating RNAPI elongation, and promote efficient rRNA processing. In this study, Seb1 allele affecting RNAPII binding have no effect on RNAPI pausing and rRNA processing. Therefore, Seb1 function in RNAPI pausing is novel and conclusion are fully supported by data presented. Seb1 study might open the way for more mechanistic study on RNAPI elongation regulation. Authors have successfully addressed all my comments (see below).

Major comments

RNAPI activity is exquisitely regulated according to growth rate. Authors have now shown that pac1 depletion is not affecting Pol I recruitment.

Authors have shown using mutant of Seb1 that RNA is required for RNAPI interaction.

Reviewer #3 (Remarks to the Author):

The revised manuscript from Duval et al., addresses the reviewers' concerns in a satisfactory manner. While not many new experiments have been performed, the authors have adjusted relevant phrases and conclusions and adjusted the figures for easier interpretation. Two small concerns remained and might require additional corrections and adjustments. Overall, I recommend this paper for publication in Nature Communication.

1. The authors clearly state in their letter to the reviewers that they did not directly measure RNAPI speed or in turn transcriptional elongation rates. Therefore, the phrasing on p.11 (lines 12-16, 26-30) and p.14 (lines 391-392) should be further adjusted to clarify that. Also, Figure S5D now includes an inference of RNAPI elongation rates based on an additional calculation. While the elongation rate is most likely impacted by the Seb1 pausing, the measurement here appears crude and distracts from

otherwise very convincing data. If the authors would like to discuss elongation rates, a technique measuring this directly would be required (see comments from reviewer 1 and 3).

2. While the authors included data to prove that the seb1 anchor away technique works efficiently in this system (Fig. S2C), the authors do not provide a control that the thiamine-induced seb1 expression repression works. It would be helpful (reviewer 3, comment 3) to include, as a minimal control, qPCR or if possible Western blotting results to show that seb1 is reduced in these experiments too.

Reviewer #4 (Remarks to the Author):

Authors have successfully addressed all my previous concerns regarding the proteomics experiments mainly focused on expanding the methods section including a LC-MS/MS subsection. The new section is very detailed, and provides enough information to replicate the proteomics experiments performed in this study. Thus, considering the proteomics experimental part, the manuscript is ready for its publication in Nature Communications.

Reviewer #1

The authors have attempted to address most of the comments raised in the previous round of review, with a notable exception of including kinetic analyses to support their conclusions. However, the revised manuscript seems to put less emphasis of transcription elongation rate versus pausing, which may be appropriate.

As noted by multiple reviewers, it is difficult to deconvolute direct and indirect effects of Seb1 on RNA polymerases I and II. The authors added a discussion of this fact, and they recognize limitations of the genetic and molecular tools employed.

In summary, the work is interesting and will generate meaningful conversation in the field.

RESPONSE: We thank the Reviewer for the positive assessment.

Reviewer #2

The revised manuscript entitled “The conserved RNA-binding protein Seb1 promotes cotranscriptional ribosomal RNA processing by controlling RNA polymerase I progression” by Maxime Duval et al establish a novel function of Seb1 in controlling rRNA production in vivo. Authors could show a clear binding to rDNA and rRNA of Seb1 using respectively Chip-seq and CRAC.

Altogether, this revised work clearly demonstrate a novel function of Seb1 in regulating RNAPI elongation, and promote efficient rRNA processing. In this study, Seb1 allele affecting RNAPII binding have no effect on RNAPI pausing and rRNA processing. Therefore, Seb1 function in RNAPI pausing is novel and conclusion are fully supported by data presented. Seb1 study might open the way for more mechanistic study on RNAPI elongation regulation. Authors have successful addressed all my comments (see below).

Major comments

RNAPI activity is exquisitely regulated according to growth rate. Authors have now shown that pac1 depletion is not affecting Pol I recruitment.

Authors have shown using mutant of Seb1 that RNA is required for RNAPI interaction.

RESPONSE: We thank the Reviewer for the positive assessment.

Reviewer #3

The revised manuscript from Duval et al., addresses the reviewers' concerns in a satisfactory manner. While not many new experiments have been performed, the authors have adjusted relevant phrases and conclusions and adjusted the figures for easier interpretation. Two small concerns remained and might require additional corrections and adjustments. Overall, I recommend this paper for publication in Nature Communication.

RESPONSE: We thank the Reviewer for the positive assessment.

1. The authors clearly state in their letter to the reviewers that they did not directly measure RNAPI speed or in turn transcriptional elongation rates. Therefore, the phrasing on p.11 (lines

12-16, 26-30) and p.14 (lines 391-392) should be further adjusted to clarify that. Also, Figure S5D now includes an inference of RNAPI elongation rates based on an additional calculation. While the elongation rate is most likely impacted by the Seb1 pausing, the measurement here appears crude and distracts from otherwise very convincing data. If the authors would like to discuss elongation rates, a technique measuring this directly would be required (see comments from reviewer 1 and 3).

RESPONSE: Although we did not directly measure RNAPI speed in this study, we did examine RNAPI elongation kinetics by analyzing the distribution of RNAPI at the single nucleotide resolution by CRAC, which measures the efficiency of RNAPI progression along the rDNA. Accordingly, and as suggested by the reviewer, we have adjusted the phrasing on p. 11 and p. 14 of the revised manuscript by replacing “elongation rates” by “transcription progression”, which describes more accurately the conclusions generated by the CRAC analysis. Regarding the data presented in Supplementary Fig. S5d of the revised manuscript, we argue that this analysis acts as an indicator of RNAPI pausing or occupancy time, rather than elongation rate. The analysis of occupancy time shown in Supplementary Fig. S5d is based on previously established RNAPI velocity in yeast (Mol. Cell 2010, 37: 809) and does not draw conclusions on RNAPI speed/elongation rate, but indicates decreased dwell time (pausing) for RNAPI over the 5.8S and ITS2 region in Seb1-deficient cells. We think this analysis is thorough and strengthens the conclusion that Seb1 slows down the progression of RNAPI during rDNA transcription.

2. While the authors included data to prove that the seb1 anchor away technique works efficiently in this system (Fig. S2C), the authors do not provide a control that the thiamine-induced seb1 expression repression works. It would be helpful (reviewer 3, comment 3) to include, as a minimal control, qPCR or if possible Western blotting results to show that seb1 is reduced in these experiments too.

RESPONSE: As suggested by the reviewer, we have now added RT-qPCR data showing thiamine-dependent reduction in *seb1* mRNA expression using the previously characterized *nmt1-seb1* conditional strain. This data is now described on p. 7 (lines 29-30) and presented on Supplementary Fig. S2a of the revised manuscript.

Reviewer #4

Authors have successfully addressed all my previous concerns regarding the proteomics experiments mainly focused on expanding the methods section including a LC-MS/MS subsection. The new section is very detailed, and provides enough information to replicate the proteomics experiments performed in this study. Thus, considering the proteomics experimental part, the manuscript is ready for its publication in Nature Communications.

RESPONSE: We thank the Reviewer for the positive assessment.